# IFITM3 restricts virus-induced inflammatory cytokine production by limiting Nogo-B mediated TLR responses

M. Clement [1,9], J. L. Forbester[1,2,9], M. Marsden[1], P. Sabberwal[1], M. S. Sommerville[1], D. Wellington [2,3], S. Dimonte[1], S. Clare[4], K. Harcourt[4], Z. Yin [2,3], L. Nobre[5], R. Antrobus[5], B. Jin[6], M. Chen[7], S. Makvandi-Nejad[2], J. A. Lindborg[8], S. M. Strittmatter [8], M. P. Weekes [5], R. J. Stanton [1], T. Dong[2,3] & I. R. Humphreys [1] ✉

Interferon-induced transmembrane protein 3 (IFITM3) is a restriction factor that limits viral pathogenesis and exerts poorly understood immunoregulatory functions. Here, using human and mouse models, we demonstrate that IFITM3 promotes MyD88-dependent, TLR-mediated IL-6 production following exposure to cytomegalovirus (CMV). IFITM3 also restricts IL-6 production in response to influenza and SARS-CoV-2. In dendritic cells, IFITM3 binds to the reticulon 4 isoform Nogo-B and promotes its proteasomal degradation. We reveal that Nogo-B mediates TLR-dependent pro-inflammatory cytokine production and promotes viral pathogenesis in vivo, and in the case of TLR2 responses, this process involves alteration of TLR2 cellular localization. Nogo-B deletion abrogates inflammatory cytokine responses and associated disease in virus-infected IFITM3-deficient mice. Thus, we uncover Nogo-B as a driver of viral pathogenesis and highlight an immunoregulatory pathway in which IFITM3 fine-tunes the responsiveness of myeloid cells to viral stimulation.

Interferon (IFN) induced transmembrane protein-3 (IFITM3) plays a major role in antiviral cellular defence, directly limiting cellular entry of a number of pathogenic viruses, including influenza A virus (IAV), human immunodeficiency virus (HIV), vesicular stomatitis virus and SARS-CoV[1–4]. Some studies have demonstrated that IFITM3, along with other members of the IFITM family of proteins, may decrease cell membrane fluidity, possibly affecting viral fusion[5,6]. In one study, alteration in membrane fluidity was attributed to upregulation of cellular cholesterol[7]. However, subsequent studies have failed to demonstrate a mechanistic link between cholesterol levels and IFITM3 activity[5,8,9]. IFITM3 has also been shown to act downstream of viral attachment, endocytosis and subsequent viral hemifusion, directly restricting full viral fusion and trapping virus particles in endosomes[10]. IFITM3 may also prevent virus entry by altering rates of virus-endosome fusion and/or accelerating the trafficking of endosomal cargo to lysosomes for destruction[11]. Furthermore, IFITM3 and other IFITM proteins can directly interact with viral proteins such as HIV-1 Env and inhibit their processing, restricting virus fusion with target

[1]Division of Infection and Immunity/Systems Immunity University Research Institute, Cardiff University, Cardiff CF14 4XN, UK. [2]MRC Human Immunology Unit, MRC Weatherall Institute of Molecular Medicine, Radcliffe Department of Medicine, Oxford University, Oxford OX3 9DS, UK. [3]Chinese Academy of Medical Sciences (CAMS) Oxford Institute (COI), University of Oxford, Oxford, UK. [4]Wellcome Sanger Institute, Wellcome Genome Campus, Hinxton, Cambridge CB10 1SA, UK. [5]Cambridge Institute for Medical Research, University of Cambridge, Hills Road, Cambridge CB2 0XY, UK. [6]Fourth Military Medical University, Xian, China. [7]Department of Microbial Pathogenesis, Yale University School of Medicine, New Haven, CT 06536, USA. [8]Departments of Neurology and Neuroscience, Yale University School of Medicine, New Haven, CT 06520, USA. [9]These authors contributed equally: M. Clement, J. L. Forbester. ✉e-mail: HumphreysIR@cardiff.ac.uk

host cell membranes[12]. Thus, IFITM3 can exert antiviral effects by influencing a broad range of cellular mechanisms.

Genetic polymorphisms within the IFITM3 locus have been linked to increased pathogenesis of viral infections such as IAV and SARS-CoV-2[13,14]. IFITM3 SNP rs34481144, which is located in the IFITM3 promoter region, is associated with lower IFITM3 mRNA levels and has a strong association with disease severity in influenza cohorts. The substitution of the major G allele with the minor C allele reduces IFITM3 expression levels, with reduced number of CD8[+] T cells in nasal washes of IAV-infected individuals[15]. The CC genotype of the rs12252 SNP, located in exon 1 of IFITM3, has been associated with increased severity of IAV infections in multiple studies[1,16].

The implication from human genetic studies is that increased viral pathogenesis in individuals with reduced IFITM3 abundance and/or function reflects only reduced control of viral entry into cells. Interestingly, however, severe IAV disease in individuals with the CC genotype of the rs12252 SNP is associated with high CCL2 levels that drive pathogenic monocyte responses and hypercytokinemia characterised by elevated levels of cytokines, including IL-6[1,17]. This hypercytokinemia is associated with fatal H7N9 infection in individuals with the rs12252 CC genotype[18]. Similarly, in mouse models of viral infection, Ifitm3 deficiency alters cytokine and chemokine profiles and leucocyte influx to sites of infection[1,19–22]. Importantly, in the murine cytomegalovirus (MCMV) model of infection, Ifitm3 restricts infection-induced IL-6 mediated viral disease, lymphopenia and loss of NK and T cells without directly impacting CMV replication[23]. In Sendai virus infection, IFITM3 inhibits IFN-β induction by promoting degradation of IRF3 within IFITM3-associated autophagosomes[24]. Moreover, mice lacking Ifitm1-3 produce elevated pro-inflammatory cytokines in response to chronic Poly I:C stimulation[25]. Thus, the biology of how IFITM3 limits viral pathogenesis may be complex and requires a better understanding. Viral entry of CMV is not susceptible to IFITM3-mediated restriction[23,26]. Thus, CMV is a tractable infectious model enabling dissection of the immune regulatory properties of IFITM3 independently of any direct antiviral restriction properties.

Herein, we investigated how IFITM3 regulates virus-induced inflammatory responses using a combination of mice and human models of CMV infection. We reveal that IFITM3 limits Toll-like receptor (TLR)-driven viral pathogenesis and that its anti-inflammatory properties are intrinsically linked with its ability to regulate the reticulon 4 isoform, Nogo-B. Using human TLR2 as a model, we show that IFITM3-Nogo-B interactions alter TLR dynamics post-viral exposure, highlighting a potential mechanism for IFITM3-Nogo-B regulation of the inflammatory response during viral infection. Overall, these data reveal a mechanism through which IFITM3 inhibits virus-induced inflammation independently of its established antiviral function and highlight the importance of Nogo-B in promoting virus-induced inflammatory responses.

## Results

### Ifitm3 regulates MyD88-dependent TLR-mediated cytokine production following MCMV infection

We have previously demonstrated that Ifitm3 restricts IL-6 mediated MCMV pathogenesis in vivo, which was associated with reduced cytokine production by DCs but was not accompanied by any defect in the direct control by Ifitm3 of virus replication[23]. To first provide evidence for a DC-intrinsic role for Ifitm3 in limiting viral disease, we irradiated wt mice and transferred bone marrow from zDC-DTR mice that enable conventional DC (cDC), depletion[27] at a 50:50 ratio with Ifitm3[−/−] bone marrow. Mice were treated (or not) with diphtheria toxin (DT) to generate mice that lacked Ifitm3 expression by DCs. Mice lacking Ifitm3[+] cDCs (DT-depleted mice) demonstrated increased weight loss (Fig. 1a) compared with mice with intact Ifitm3[+] DCs. Exacerbated weight loss in mice lacking Ifitm3 expression by DCs was accompanied by elevated IL-6 production (Fig. 1b) without any impact

on virus load (Fig. 1c), thus suggesting the importance of Ifitm3 expression by DCs in MCMV pathogenesis.

Ifitm3 limits TLR-induced cytokine production by bone marrow-derived DCs (BM-DCs[23]), a result that we reproduced with a broad panel of TLR ligands, with the exception of TLR3 or TLR5 stimulation which induced low IL-6 production in these experiments (Fig. 1d). Endosomal TLR3, TLR7 and TLR9 are important in the recognition of MCMV and subsequent activation of innate immune responses[28,29]. Thus, to understand whether endosomal TLRs mediated MCMV-induced IL-6 production in DCs, Tlr3[−/−], Tlr7[−/−] and Tlr9[−/−] BM-DCs were infected with MCMV and IL-6 was measured (Fig. 1e). Tlr7[−/−] and Tlr9[−/−] BM-DCs were particularly defective in IL-6 production, suggesting a dominant role for these pattern recognition receptors in MCMV-induced pro-inflammatory cytokine production. In accordance, blocking Tlr7 (Fig. 1f) and Tlr9 (Fig. 1g) signalling significantly reduced IL-6 in both wt and Ifitm3[−/−] cells to similar levels following infection with MCMV, suggesting that Ifitm3-mediated regulation of these responses restricts MCMV-induced cytokine production.

The adaptor protein MyD88 is downstream of various TLRs, including TLR7 and TLR9. To provide evidence for a role of TLR signalling in Ifitm3-regulated viral pathogenesis, we crossed Ifitm3[−/−] and Myd88[−/−] mice and infected them with MCMV. Genetic deficiency of MyD88 in Ifitm3[−/−] mice resolved infection-induced weight loss to wt levels (Fig. 1h), indicating that signalling through MyD88 is required to drive exacerbated weight loss in Ifitm3[−/−] mice. Generation of BM-DCs from Ifitm3[−/−]/Myd88[−/−] mice confirmed that IL-6 production post-MCMV exposure was reduced in Ifitm3[−/−]/Myd88[−/−] double knockout DCs in comparison to wt Ifitm3/Myd88 (Fig. 1i). MyD88 also acts downstream of IL-1R[30]. However, antagonising IL-1R signalling with anakinra had no impact on MCMV weight loss (Supplementary Fig. 1a) or IL-6 production (Supplementary Fig. 1b). Thus, these data suggest that MyD88-dependent TLR signalling drives enhanced pathogenesis in Ifitm3[−/−] mice.

### IFITM3 restricts HCMV-induced TLR2-mediated cytokine production by human DCs

Next, we sought to investigate whether IFITM3 regulated virus-induced cytokine production in human DCs. We generated DCs from healthy control (Kolf2) iPSCs, and iPSCs with biallelic mutations in IFITM3 generated in the Kolf2 background using CRISPR/Cas9 engineering[31]. Two clones (IFITM3[−/−] F01 and IFITM3[−/−] H12) were selected for downstream assays, to control for risk of off-target mutations. Kolf2, IFITM3[−/−] H12 and IFITM3[−/−] F01 were differentiated into iPS-DCs using previously published methodology[32]. DC morphology in culture was similar for all three lines (Supplementary Fig. 2a), and IFITM3 deficiency did not affect the efficiency of DC differentiation, with similar numbers of DC precursors harvested from each line (Supplementary Fig. 2b), and similar surface expression of DC markers CD11c and CD141 (Supplementary Fig. 2c). Furthermore, as IFITM1 and IFITM2 have significant sequence homology with IFITM3, we determined that mRNA (Supplementary Fig. 2d) and protein (Supplementary Fig. 2e) levels of IFITM1 and IFITM2 were not significantly decreased in iPSCs in comparison to Kolf2 control cells in either IFITM3[−/−] clone used in this study. After differentiation into iPS-DCs and challenge with HCMV (Merlin strain), IFITM3 protein was detected only in Kolf2 iPS-DCs and not in either IFITM3[−/−] line, using a previously validated IFITM3-specific antibody[16,31] (Fig. 2a). In accordance with data generated in the mouse/MCMV system, IL-6 responses were elevated after HCMV stimulation of iPS-DCs that lacked IFITM3 (Fig. 2b). iPS-DCs were non-permissive to productive HCMV infection after low multiplicity of infection (MOI = 5), irrespective of IFITM3 expression (Supplementary Fig. 3a, b). Thus, IFITM3 suppresses IL-6 production in DCs to both HCMV and MCMV without impacting virus entry.

Analogous to data in murine cells, IL-6 secretion was enhanced in IFITM3-deficient iPS-DCs following stimulation of TLR2, TLR3, TLR4

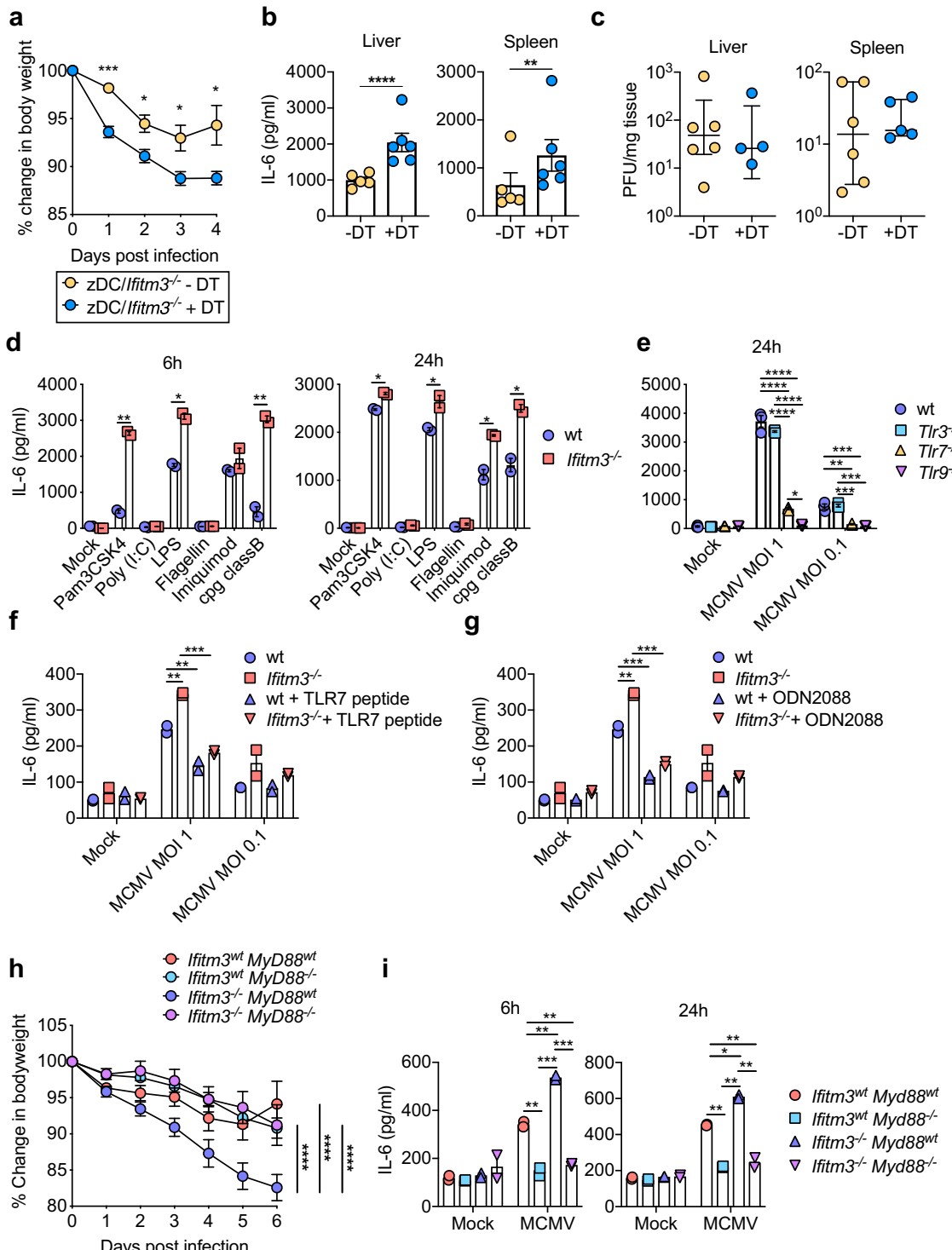

**Fig. 1 | Ifitm3 enhances IL-6 downstream of TLRs and MyD88.** Female mixed bone marrow chimeras with 50:50 of *Ifitm3⁻/⁻* and wt-zDC-DTR were generated and treated (*n* = 6) or not (*n* = 5) with DT. **a** Mice were infected with 5 × 10⁵ PFU MCMV and weight loss was assessed over time. Mean and SEM are shown. **b** 4 days p.i. spleens and livers were harvested from MCMV-infected mice, homogenised, and IL-6 was assayed. Mean and SEM are shown. **c** Replicating virus in liver and spleen was quantified 4 days p.i. by plaque assay. Individual data points, median and inter-quartile range are shown. **d** BM-DCs from wt and *Ifitm3⁻/⁻* mice were stimulated with TLR ligands and IL-6 in supernatants was assayed 6 and 24 h post stimulation. **e** BM-DCs from wt, *Tlr3⁻/⁻*, *Tlr7⁻/⁻* and *Tlr9⁻/⁻* mice were infected with MCMV (MOI 1 or 0.1), and IL-6 in supernatants was assayed 6 and 24 h post infection. **f** BM-DCs from wt and *Ifitm3⁻/⁻* mice were pre-incubated with or without TLR7 synthetic blocking

peptide or **g** ODN 2088 for 1 h prior to infection with MCMV (MOI 1 or 0.1). IL-6 in supernatants was assayed 24 h post infection. **d**–**g** Data are shown as mean ± SEM. **h** *Ifitm3ᵂᵗMyD88ᵂᵗ* (*n* = 8), *Ifitm3⁻/⁻MyD88ᵂᵗ* (*n* = 7), *Ifitm3ᵂᵗMyD88⁻/⁻* (*n* = 7), *Ifitm3⁻/⁻MyD88⁻/⁻* (*n* = 6) male and female mice (mixed genders in all groups) were infected with 5 × 10⁵ PFU MCMV and weight loss was assessed over time. Data are shown as mean ± SEM. **i** BM-DCs from *Ifitm3ᵂᵗMyD88ᵂᵗ*, *Ifitm3⁻/⁻MyD88ᵂᵗ*, *Ifitm3ᵂᵗMyD88⁻/⁻*, *Ifitm3⁻/⁻MyD88⁻/⁻* mice were infected with MCMV (MOI 1), and IL-6 in supernatants was assayed 6 and 24 h post infection. Data are shown as mean ± SEM. Statistical significance was assessed using Student's *t* test (**a**, **b**, **d**), Mann–Whitney *U* (**c**), one-way ANOVA analysis with Tukey's multiple comparisons test (**e**–**g**, **i**) or two-way ANOVA analysis (**h**). *p* values are reported as follows: n.s., >0.05; *, ≤0.05; **, ≤0.01; ***, ≤0.001; and ****, ≤0.0001. Source data are provided as a Source Data file.

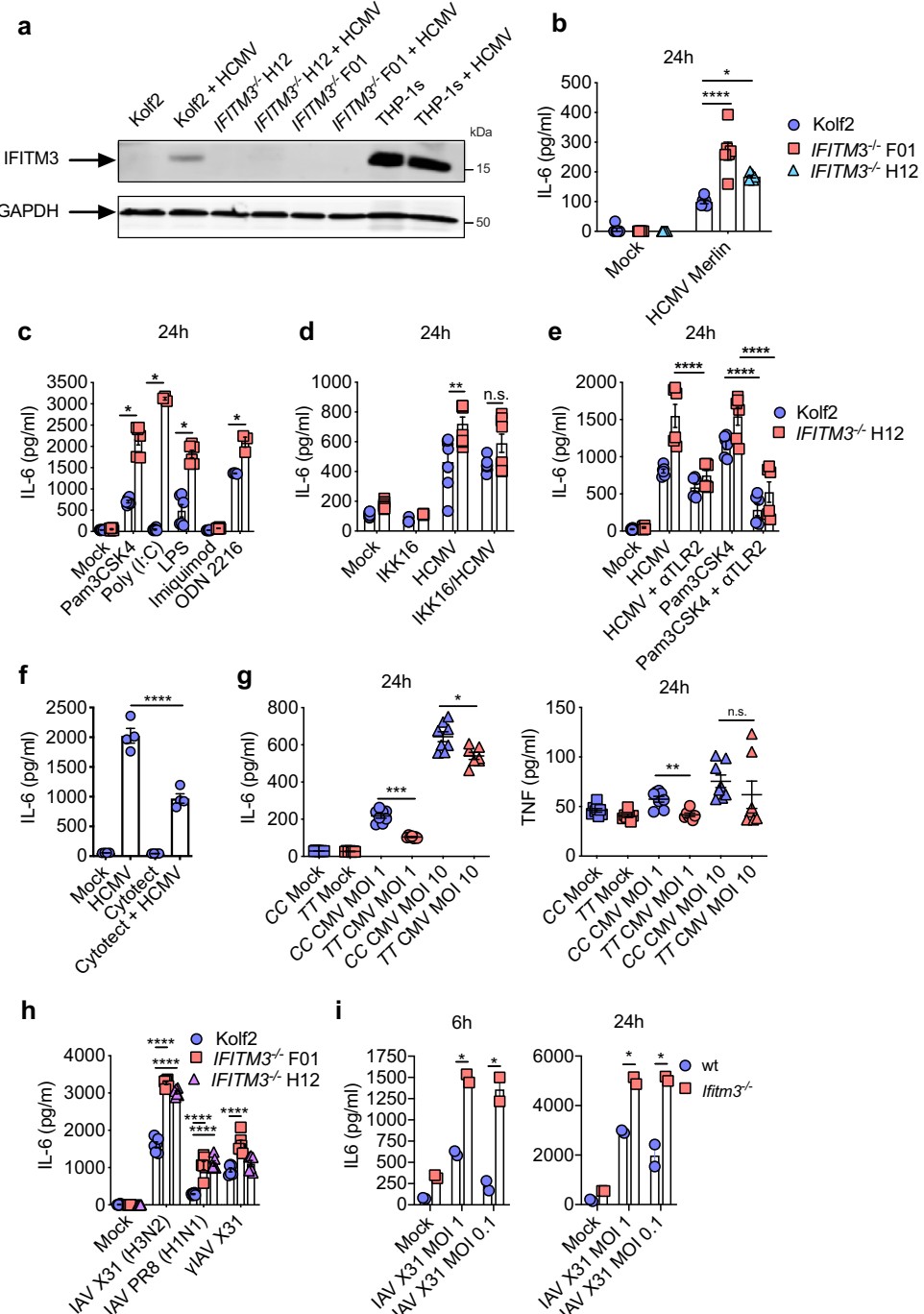

**Fig. 2 | IFITM3 regulates HCMV- and TLR-induced IL-6 in human DCs.** ELISA data presented from assays performed in triplicate or more, for at least two independent technical replicates per assay. HCMV (merlin strain) was used for all iPS-DC experiments at MOI 5 unless otherwise stated. **a** IFITM3 or GAPDH protein levels after stimulation of control and *IFITM3⁻/⁻* iPS-DCs for 24 h with HCMV were measured by Western blot. Lysate from THP-1 cells was used as a positive control. **b** iPS-DCs (*n* = 3–6 separate cell cultures in different wells) were stimulated with HCMV, and IL-6 in supernatants were assayed 24 h later. **c** iPS-DCs (*n* = 3–6 separate cell cultures in different wells) were stimulated with TLR ligands and IL-6 was measured. **d** iPS-DCs (*n* = 4–8 separate cell cultures in different wells) were pre-treated with or without IKK16 for 1 h and then stimulated with HCMV and IL-6 measured in supernatants. **e** iPS-DCs (*n* = 6 separate cell cultures in different wells) were pre-treated for 1 h with or without neutralising antibody to TLR2 and stimulated with HCMV or TLR2 ligand Pam3CSK4, and IL-6 was assayed after 24 h. **f** iPS-DCs

(*n* = 4 separate cell cultures in different wells) were pre-treated with Cytotect CP Biotest, or left untreated, and stimulated with HCMV, with IL-6 in supernatant assayed 24 h later. **g** Monocyte-derived DCs isolated from human donors genotyped for SNP rs12252 (*n* = 7–8 different donors) were stimulated with HCMV (MOI 1 or 10), and IL-6 and TNF in supernatants were assayed 24 h post infection. **h** iPS-DCs (*n* = 6 separate cell cultures in different wells) were stimulated with IAV A/X31 (H3N2), PR8 (H1N1), or gamma-irradiated A/X31 (MOI 1) and IL-6 was measured 6 and 24 h later. **i** BM-DCs from wt and *Ifitm3⁻/⁻* mice (*n* = 2 separate cell cultures in different wells) were infected with IAV A/X31 (H3N2) (MOI 1 or 0.1), and IL-6 in supernatants was assayed 6 and 24 h later. Mean ± SEM are shown, and statistical significance was assessed using one-way ANOVA analysis with Tukey's multiple comparisons test (**b**, **d**–**f**, **h**) or Student's *t* test (**c**, **g**, **i**). *p* values are reported as follows: n.s., >0.05; *, ≤0.05; **, ≤0.01. Source data are provided as a Source Data file.

and TLR9 (Fig. 2c). Antagonising NFκB using the inhibitor IKK-16, which targets IκB kinase (IKK)[32], reduced IL-6 in *IFITM3*[-/-] iPS-DCs closer to WT levels, suggesting enhanced IL-6 in *IFITM3*-deficient cells is partially NFκB dependent (Fig. 2d). This NFκB inhibition had no effect on IL-6 in WT cells, suggesting that in cells expressing IFITM3 IL-6 production is not NFκB-dependent. TLR2 is important for triggering the inflammatory cytokine response to HCMV[33] following binding to viral surface glycoproteins gB and gH[34]. Neutralising antibody to TLR2 significantly reduced the IL-6 response that was otherwise amplified in *IFITM3*[-/-] iPS-DCs following exposure to either HCMV or a TLR2 ligand (Fig. 2e). Furthermore, pre-incubation with Cytotect (pooled hyperimmune globulin against HCMV) also reduced IL-6 significantly, providing further evidence for the role of TLR2 in HCMV glycoprotein recognition in mediating subsequent IL6 signalling (Fig. 2f). These data thus suggested that in human DCs TLR2-mediated recognition of HCMV is particularly important for driving the inflammatory response by these cells, and that this inflammatory process was limited by IFITM3. Finally, we generated primary blood monocyte-derived DCs (mDCs) from human donors genotyped for the *IFITM3* SNP rs12252 and stimulated cells with HCMV, with these cells being non-permissive to HCMV infection similarly to iPS-DCs (Supplementary Fig. 3c). Donors with the *CC* allele that associate with reduced IFITM3 function[17] also demonstrated increased IL-6 and TNF responses following viral stimulation (Fig. 2g), suggesting that variation within the human IFITM3 locus could influence differential cytokine responses by myeloid cells in humans.

## IFITM3 regulates DC inflammatory cytokine production in response to evolutionarily diverse pathogenic viruses

*Ifitm3*[-/-] mice are more susceptible to IAV infection, exhibiting loss of viral control, but also alterations in immune responses[1], and variation in human *IFITM3* has been associated with more severe disease after IAV[1] and SARS-CoV-2 infection[14]. This suggests that Ifitm3 may regulate pro-inflammatory cytokine responses in response evolutionarily divergent viruses independently of control of virus entry into cells. We therefore first stimulated iPS-DCs with SARS-CoV-2 (strain Victoria 01/20). IFITM3 has pro- and antiviral effects on SARS-CoV-2 infection depending on the cell type studied[35–37]. Importantly, in vivo data suggest that Iftim3 protects from viral pathogenesis and that this is associated with elevated pro-inflammatory cytokine production[38]. In accordance, we observed elevated IL-6 in *IFITM3*[-/-] DCs, although this was not statistically significant ($p = 0.07$) (Supplementary Fig. 3d).

We have previously shown in iPS-DCs, IAV-induced IL-6 occurs downstream of sensing by TLR7 and possibly RIG-I[32]. Interestingly, when we infected human iPS-DCs with IAV H1N1 (PR8) and H3N2 (X31), we observed no direct IAV restriction (Supplementary Fig. 3e), but significantly enhanced IL-6 was observed in response to infectious and γ-irradiated virus (Fig. 2h). In accordance to observations in human DCs, enhanced IL-6 was also observed in murine *Ifitm3*[-/-] BM-DCs in response to IAV (Fig. 2i). Therefore, these data suggest that enhancement of viral induced IL-6 in IFITM3-deficient cells is independent of the viral restriction role IFITM3 plays in other cell types. Thus, overall, our data reveal that IFITM3 regulates NFκB, MyD88 and TLR-dependent virus-induced pro-inflammatory cytokine production and that this is potentially relevant in evolutionarily diverse viruses.

## IFITM3 binds the TLR pathway-associated protein Nogo-B

To determine the mechanism(s) by which Ifitm3 limits TLR-mediated cytokine production, we used a proteomic approach to screen for Ifitm3 binding partners. We performed two separate immunoprecipitation (IP) experiments from cells cultured with differentially labelled amino acids (stable isotype labelling of cells in culture (SILAC)-IP, Supplementary Data 1). Both experiments used an Ifitm3-specific

antibody to pull down Ifitm3, along with any binding partners, from either wt ('Medium' labelled) or *Ifitm3*[-/-] ('Heavy' labelled) BM-DCs. In one condition (Fig. 3a) cells were mock infected while in the other (Fig. 3b) cells were infected with MCMV. Analysis of the ratio of peptides from specific proteins in each condition revealed enrichment of receptor enhancing expression protein 5 (Reep5); Ifitm1 and 2; PRA1 family protein 3 (Arl6ip5, Praf3) and members of the reticulon family of proteins Rtn3 and Rtn4 (Nogo) (Fig. 3a, b).

Interactions between IFITM3 and Nogo (but not Nogo binding to IFITM1 or IFITM2) have previously been reported in interactome studies[39,40]. The *Rtn4* gene encodes several alternatively spliced transcript variants, with the isoforms encoding members of the reticulon family of proteins. The main isoforms are Nogo-A, mainly expressed in the nervous system, Nogo-B, which is ubiquitously expressed in most cell types, and Nogo-C, enriched in skeletal muscle[41]. Nogo-B is thought to be involved in maintaining endoplasmic reticulum (ER) shape[42]. Intriguingly, however, a regulatory role[43] of Nogo-B in TLR trafficking and expression has been identified[43,44]. Interestingly, whereas blotting for Nogo-A of iPS-DC lysates confirmed no Nogo-A expression by myeloid cells (Supplementary Fig. 4a), we confirmed IFITM-3-Nogo-B interaction in BM-DCs by IP-western blot (Fig. 3c), thus demonstrating that Ifitm3 binds Nogo-B in dendritic cells prior to and following CMV exposure. Furthermore, immune fluorescence demonstrated Rab7[+] endosomes but not Lamp1[+] lysosomes as one subcellular location of IFITM3-Nogo-B co-localisation (Supplementary Fig. 4b).

## Nogo-B promotes viral pathogenesis

We hypothesised that the interaction between Ifitm3 and Nogo-B was relevant for the immune-regulatory function of Ifitm3. Importantly, MCMV-infected *Nogo-A/B*[-/-] mice exhibited significantly reduced weight loss in comparison to wt mice (Fig. 3d), despite no influence on viral load (Fig. 3e), which was accompanied by reduced pro-inflammatory cytokine production in virus-infected tissue (Fig. 3f, g). Furthermore, IL-6 production by *Nogo-A/B*[-/-] BM-DCs was significantly reduced following stimulation with TLR2, 4, 7 and 9 ligands (Fig. 3h), and also in response to MCMV and IAV X31 (Fig. 3i). Although *Nogo-A/B*[-/-] mice lack expression of both the NogoA and NogoB isoforms, we reasoned that ubiquitous expression only of the Nogo-B isoform suggested that only this Nogo isoform regulates TLR and virus-induced cytokine production and in vivo viral pathogenesis.

We investigated whether IFITM3 limited viral inflammation by altering Nogo-B abundance. Firstly, we quantified Nogo-B levels post-HCMV exposure in human iPS-DCs and observed a significant enhancement of Nogo-B levels in IFITM3-deficient iPS-DCs at early time-points (1 h and, to a lesser extent, 3 h, Fig. 4a, b), suggesting that elevated pro-inflammatory cytokine production in IFITM3-deficient cells may reflect elevated Nogo-B activity. In accordance, using siRNAs specific for *IFITM3* and *Rtn4/Nogo* in the monocytic cell line THP-1 in which we could achieve significant knockdown of expression of both proteins (Fig. 4c), we demonstrated that, after exposure to HCMV, Nogo-B knockdown reduced HCMV-induced IL-6 secretion whereas IFITM3 knockdown exacerbated this (Fig. 4d). Importantly, knockdown of both IFITM3 and Nogo-B restored viral-induced IL-6 to level observed with control siRNA (Fig. 4d). We observed similar results in experiments using murine BM-DCs (Fig. 4e, f). These data suggest that in IFITM3-deficient DCs, dysregulated Nogo-B may contribute to enhanced inflammatory responses. However, these experiments were confounded by incomplete Nogo-B (and IFITM3) knockdown by siRNAs. Therefore, to establish complete deletion of Ifitm3 and Nogo-B gene expression, we generated *Ifitm3*[-/-] *NogoA/B*[-/-] mice and compared MCMV-induced IL-6 secretion by BM-DCs lacking Ifitm3, Nogo-B or both (Fig. 4g). Importantly, deleting Nogo-B in *Ifitm3*[-/-] BM-DCs reduced virus-induced cytokine production to levels comparable with *NogoA/B*[-/-] BM-DCs and lower than WT cells (Fig. 4g), suggesting a

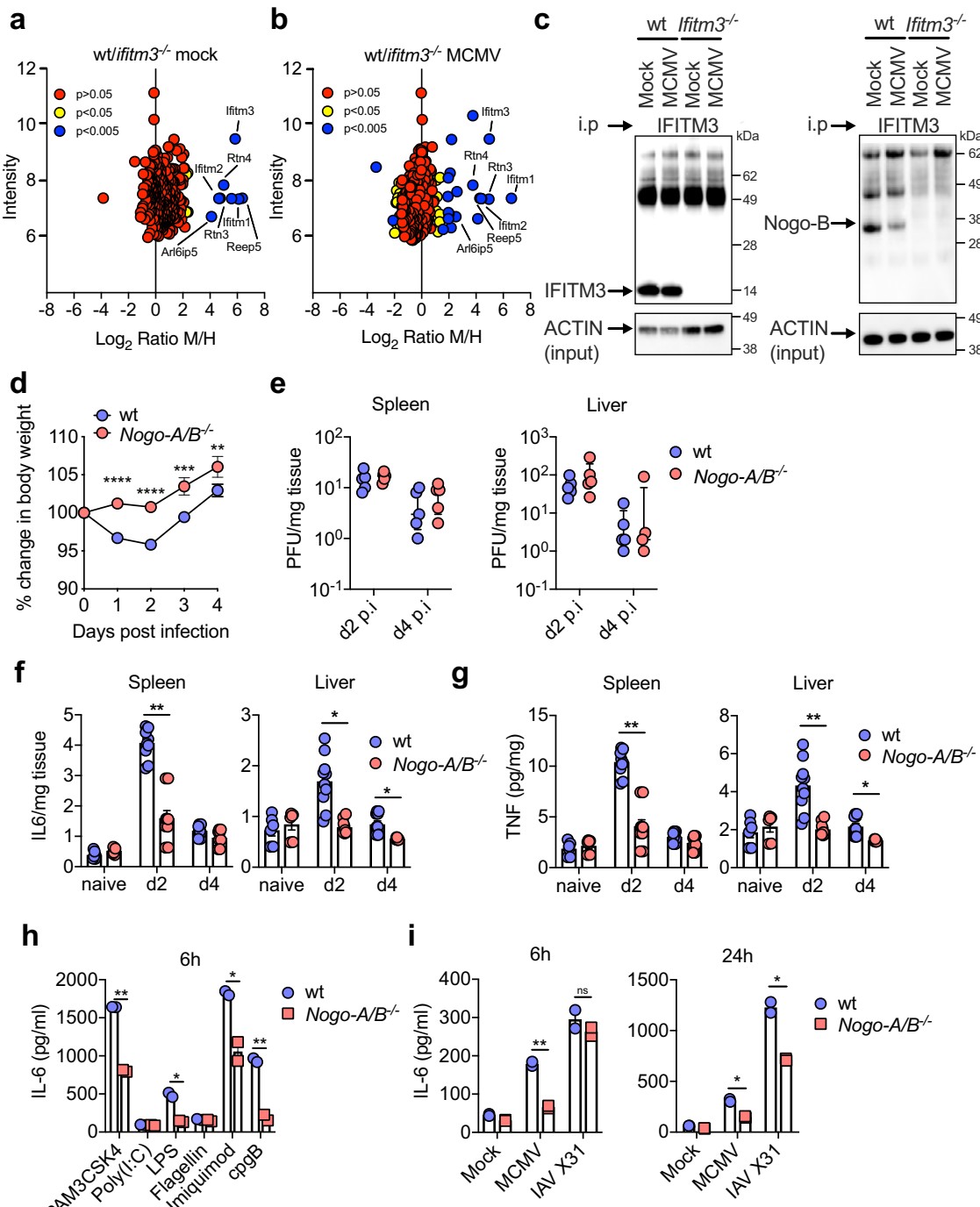

**Fig. 3 | Ifitm3 interacts with the reticulon protein Nogo-B. a, b** GM-CSF differentiated BM-DCs from wt and *Ifitm3*[-/-] mice were grown in 'Medium' or 'Heavy' SILAC medium respectively. Cells were either mock infected (**a**) or infected with MCMV (MOI 1) (**b**) for 3 h, lysed and IP for anti-fragilis (Ifitm3) was performed. The fold enrichment of each protein is shown. *p* values were estimated using significance A values (two-tailed), then corrected for multiple hypothesis testing using the Benjamini–Hochberg correction (82). **c** BM-DCs from wt and *Ifitm3*[-/-] mice were infected with MCMV (MOI 1) for 3 h, lysed and IP for anti-fragilis was performed. Ifitm3, Nogo-B and ACTIN (input samples only) levels were detected by Western blot. Data represent two separate experiments. **d** Wt and *Nogo-A/B*[-/-] male and female mice were infected with MCMV and weight loss was assessed over time. Data are shown as mean ± SEM from of 25 (wt) and 26 (*Nogo-A/B*[-/-])

mice. **e** Replicating virus from harvested spleens and livers was measured by plaque assay at d2 and d4 p.i. Individual data points (*n* = 5/group), median and inter-quartile range are shown. Harvested spleens and liver tissue supernatant from either naïve, or from d2 and d4 p.i. was assayed for (**f**) IL-6 and (**g**) TNF. Data are shown as mean ± SEM (*n* = 6–10 mice/group) and represent three replicate experiments. BM-DCs from wt and *Nogo-A/B*[-/-] mice were stimulated with or without (**h**) TLR ligands or (**i**) MCMV or IAV A/X31 (MOI 1), for 6 and 24 h and IL-6 was assayed in supernatants. Data are shown as mean ± SEM of two biologically independent cultures. Statistical significance was assessed using Student's *t* test (**d, f–i**) or Mann–Whitney *U* (**e**). *p* values are reported as follows: n.s., >0.05; *, ≤0.05; **, ≤0.01; ***, ≤0.001; and ****, ≤0.0001. Source data are provided as a Source Data file.

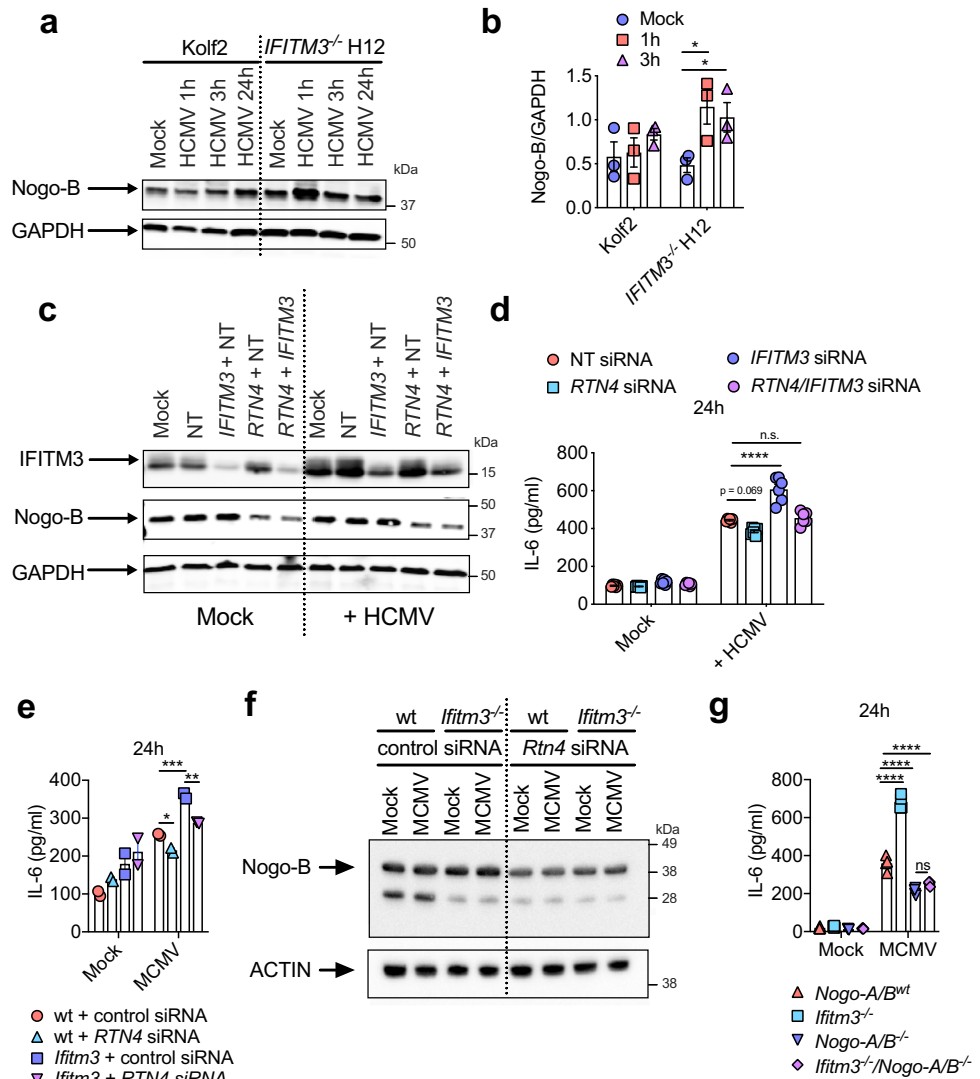

**Fig. 4 | Nogo-B/IFITM3 interaction regulates CMV-induced IL-6. a** Nogo-B and GAPDH was detected by Western blot, after stimulation of Kolf2 and *IFITM3*[-/-] H12 iPS-DCs for 1, 3 or 24 h with HCMV (MOI 5), or mock treatment, and preparation of whole-cell extracts. Data represent three experiments. **b** Relative expression of Nogo-B to GAPDH from **a** was assessed using ImageJ software. Mean ± SEM of three biological replicates are shown. **c** THP-1s were treated for 72 h with *RTN4/Nogo*, Non-targeting (NT) and/or *IFITM3* siRNAs and assayed for anti-IFITM3 and anti-Nogo-B by Western blot 72 h after siRNA addition and preparation of whole-cell extracts. Data represent two separate experiments. **d** THP-1s (*n* = 6 separate cell cultures in different wells) treated for 72 h with *RTN4/Nogo*, Non-targeting (NT) and/or *IFITM3* siRNAs were stimulated with HCMV (MOI 5) for 24 h, with IL-6 in supernatant measured. **e** BM-DCs (*n* = 2, *n* = 4 separate cell cultures in different

wells) from wt and *Ifitm3*[-/-] mice targeted with either *Rtn4/Nogo* targeting siRNA or AllStars control siRNA, and were infected with MCMV (MOI 1), and IL-6 in supernatants was assayed 6 and 24 h post infection. **f** Nogo-B expression was detected in *Rtn4/Nogo* siRNA-treated wt and *Ifitm3*[-/-] BM-DCs by Western blot at 3 h post infection with MCMV. Data represent two experiments. **g** BM-DCs (*n* = 4 separate cell cultures in different wells) from *Nogo-A/B*[wt], *Ifitm3*[-/-], *Nogo-A/B*[-/-] and *Ifitm3*[-/-]*Nogo-A/B*[-/-] mice were infected with MCMV (MOI 1) for 24 h with IL-6 assayed in supernatants. Data are shown as mean ± SEM and statistical significance was assessed using Student's *t* test for relevant comparisons (**b**) and one-way ANOVA with Tukey's multiple comparisons test (**d**, **e**, **g**). *p* values are reported as follows: n.s., >0.05; *, ≤0.05; **, ≤0.01; ***, ≤0.001; and ****, ≤0.0001. Source data are provided as a Source Data file.

direct association between Ifitm3 and Nogo-B in regulation of virus-induced cytokine production.

To demonstrate the importance of the interaction between Ifitm3 and Nogo-B in vivo, we infected single and double knockout mice with MCMV. *Ifitm3*[-/-]*NogoA/B*[-/-] mice exhibited comparable weight loss (Fig. 5a) and IL-6 (Fig. 5b) production to wt mice, clearly demonstrating the importance of Nogo-B in driving viral pathogenesis in the absence of Ifitm3. We previously demonstrated that overt IL-6 production in response to MCMV infection in *Ifitm3*[-/-] mice resulted in elevated viral load[23]. In accordance with the conclusion that Nogo-B drives this process, Nogo-B deficiency in *Ifitm3*[-/-] mice rescued the defect in control of virus replication observed in these mice (Fig. 5c).

## IFITM3 regulates Nogo-B expression in a proteasomal dependent manner

Nogo-B abundance is regulated via the proteasome[45]. Accordingly, inhibition of proteasomal cleavage using MG-132 in iPS-DCs (Fig. 6a, b) and murine BM-DCs (where Nogo-B protein was increased in Ifitm3 deficiency, Fig. 6c, d) resulted in a significant increase in Nogo-B protein in wt but not IFITM3-deficient cells in both systems. Thus, IFITM3 regulated Nogo-B turnover by promoting proteasomal degradation.

## IFITM3-Nogo-B interactions regulate TLR2 dynamics

We wanted to understand how IFITM3-Nogo-B interactions influenced virus-induced cytokine production. Given that (1) our data implied a critical role for TLRs in this process and (2) Nogo-B has been implicated

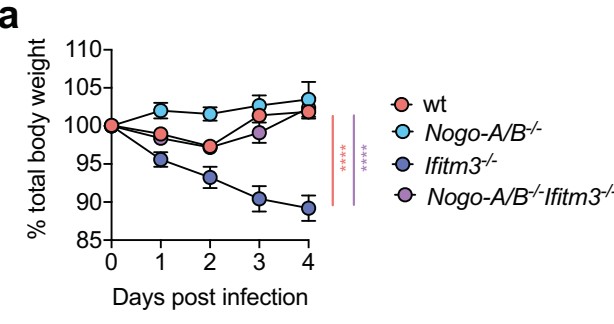

**a**

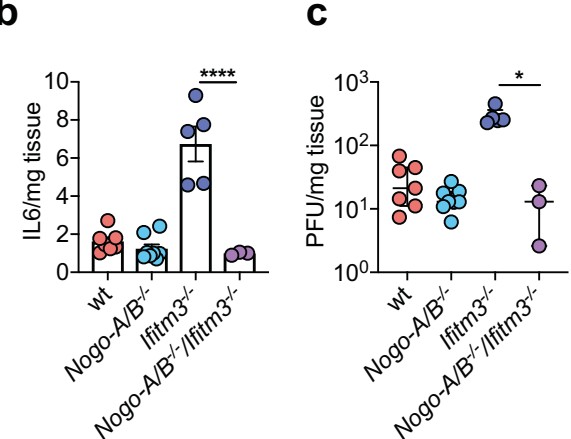

**b**

**c**

**Fig. 5 | Nogo-B drives viral pathogenesis in *Ifitm3*<sup>−/−</sup> mice during MCMV infection.** Wt ($n = 13$), *Ifitm3*<sup>−/−</sup> ($n = 9$), *Nogo-A/B*<sup>−/−</sup> ($n = 14$), and *Nogo-A/B*<sup>−/−</sup>*Ifitm3*<sup>−/−</sup> ($n = 9$) male and female mice were infected with MCMV for 4 days. Data are merged from two independent experiments. **a** Weight changes were measured over time and data are shown as mean ± SEM. Statistical significance between weight curves of *Ifitm3*<sup>−/−</sup> and either wt (black) or *Nogo-A/B*<sup>−/−</sup>*Ifitm3*<sup>−/−</sup> (blue) mice is shown. After 4 days, IL-6 (**b**) and virus load (**c**) in spleen homogenates were measured in wt ($n = 7$), *Nogo-A/B* ($n = 8$), *Ifitm3*<sup>−/−</sup> ($n = 5$) and *Nogo-A/B*<sup>−/−</sup>*Ifitm3*<sup>−/−</sup> ($n = 3$) mice. Data are representative of two experiments and are shown as mean ± SEM (**b**) or individual mice + median and inter-quartile range (**c**). Statistical significance was assessed using two-way ANOVA (**a**) or one-way ANOVA analysis with Tukey's multiple comparisons test (**b**, **c**). $p$ values are reported as follows: n.s., >0.05; *, ≤0.05; **, ≤0.01; ***, ≤0.001. Source data are provided as a Source Data file.

in impacting TLR locality within cells[43,44], we examined the impact of IFITM3 deficiency on TLR localisation. As TLR2-mediated cytokine responses were regulated by IFITM3 and TLR2 is expressed on the cell surface, we decided to use human cells and TLR2 as a model to investigate whether IFITM3 alters TLR dynamics in response to virus. We performed immunostaining in Kolf2 and *IFITM3*<sup>−/−</sup> H12 iPS-DCs to visualise TLR2 and Nogo-B expression (Fig. 7a). Our observations suggested colocalization (yellow) between TLR2 and Nogo-B after HCMV exposure (Fig. 7a and Supplementary Fig. 5); and that TLR2 and Nogo-B cellular localisation was influenced by IFITM3, with more TLR2/Nogo-B localisation visible as cytoplasmic puncta in *IFITM3*<sup>−/−</sup> iPS-DCs 24 h post infection (Fig. 7a). The distribution of most Nogo-B protein in these cells did not match a peripheral ER pattern. By flow cytometry, we observed significantly more cell surface TLR2 expression by HCMV-exposed Kolf2 iPS-DCs in comparison to mock treated cells (Fig. 7b and Supplementary Fig. 6a). In contrast, surface TLR2 expression on *IFITM3*<sup>−/−</sup> DCs does not increase after HCMV exposure and was significantly reduced when compared directly to wt DCs (Fig. 7b). Interestingly, following exposure to the TLR2 ligand Pam3CSK4, surface TLR2 expression reduced rapidly and dramatically in *IFITM3*<sup>−/−</sup> DCs 1 h post-Pam3CSK4 exposure, whereas TLR2 surface levels in wildtype Kolf2 cells only reduced later at 4 h before rising again by 6 h post-

exposure (Fig. 7c). We next used our siRNA THP-1 system to assess TLR2 dynamics post-HCMV exposure in response to Nogo and/or IFITM3 knockdown (Fig. 7d, e and Supplementary Fig. 6b). Here, we observed decreased TLR2 surface expression with IFITM3 knockdown at early time-points at 3 h post-virus exposure. By contrast, following Nogo knockdown, we observed a less dramatic reduction in surface TLR2 as compared to both non-transfected and IFITM3 siRNA-treated cells, and markedly increased TLR2 surface expression 24 h post-viral exposure. In cells where both Nogo and IFITM3 were knocked down, TLR2 surface expression was comparable to Nogo siRNA-treated cells (Fig. 7d, e).

Our data imply that Nogo-B regulates TLR2 internalisation and that this process may be regulated by IFITM3. Internalisation of TLR2 and its ligand into endosomes may be important for determining inflammatory cytokines and type I IFN responses, with several studies suggesting that TLR2 generates different signals from different locations within the endosomal pathway, depending on the specific TLR2 ligand used[46–49]. Upon ligand binding, TLR2 is rapidly internalised and trafficked to the Golgi Apparatus[50], with evidence in some cell types that signalling from the cell surface could be important for inflammatory cytokine induction whereas endosomal signalling is important for type-I IFN induction[51]. However, in human monocytes NFκB activation requires, internalisation of TLR2 into endosomal compartments[49]. Importantly, when iPS-DCs were treated with the endocytosis inhibitor ES9-17[52], HCMV-induced IL-6 was reduced only in *IFITM3*<sup>−/−</sup> cells (Fig. 7f). Furthermore, concurrent treatment of *IFITM3*<sup>−/−</sup> iPS-DCs with ES9-17 and the NFκB inhibitor IKK-16 had minimal additive effect on HCMV-induced cytokine production, suggesting that in IFITM3-deficient cells enhanced IL-6 is driven by an endocytosis and TLR2-dependent mechanism that is mediated by NFκB signalling. In contrast, treatment with Bafilomycin A1 and ammonium chloride (NH₄Cl), which inhibit the late stages of autophagy and the acidification of the late endosome, did not impact on IL-6 production in either healthy control Kolf2 or *IFITM3*<sup>−/−</sup> H12 iPS-DCs after HCMV exposure (Fig. 7g). Thus, these data imply that IFITM3 regulates virus-induced inflammatory cytokine production by influencing Nogo-B orchestrated movement of TLR that, in the case of TLR2, is an endocytosis-dependent event.

## Discussion

Herein, we demonstrate using murine and human in vitro systems that IFITM3 interactions with the reticulon family member Nogo-B represent a key mechanism for the regulation of inflammatory cytokine production induced by TLRs and evolutionary diverse viruses, and we demonstrate an important role for the Nogo-B/TLR axis in viral pathogenesis in vivo. We identified dendritic cells as the key cell type responsible for producing excess IL-6 during MCMV-induced inflammation in IFITM3-deficiency. DCs play a key role in bridging the innate and adaptive immune system[53] and the production of inflammatory cytokines by DCs requires careful balance to ensure sufficient orchestration of antiviral immunity without exacerbating pathology.

Nogo-B, one of the major isoforms of the *RTN4* gene, is a widely expressed reticulon protein that is thought to be involved in formation and stabilisation of the ER[42]. However, the role of Nogo-B in immune regulation has been recently suggested, with data indicating a role for Nogo-B in fine-tuning intracellular TLR pathways by influencing their locality within cells, certainly in the case of delivery of TLR9 to endolysosomes[44]. Furthermore, Nogo-B promotes surface expression of TLR4 and TLR-induced cytokine production in macrophages[43]. In our study, we observed co-localisation of TLR2 and Nogo-B, and knockdown of Nogo-B led to alterations in TLR2 movement within the cell after viral stimulation. Thus, it is possible that Nogo-B influenced virus-induced cytokine production by directly altering TLR movement. However, TLR2 is internalised following activation[54] and altered TLR2 dynamics in our experiments may reflect alterations in initial activation

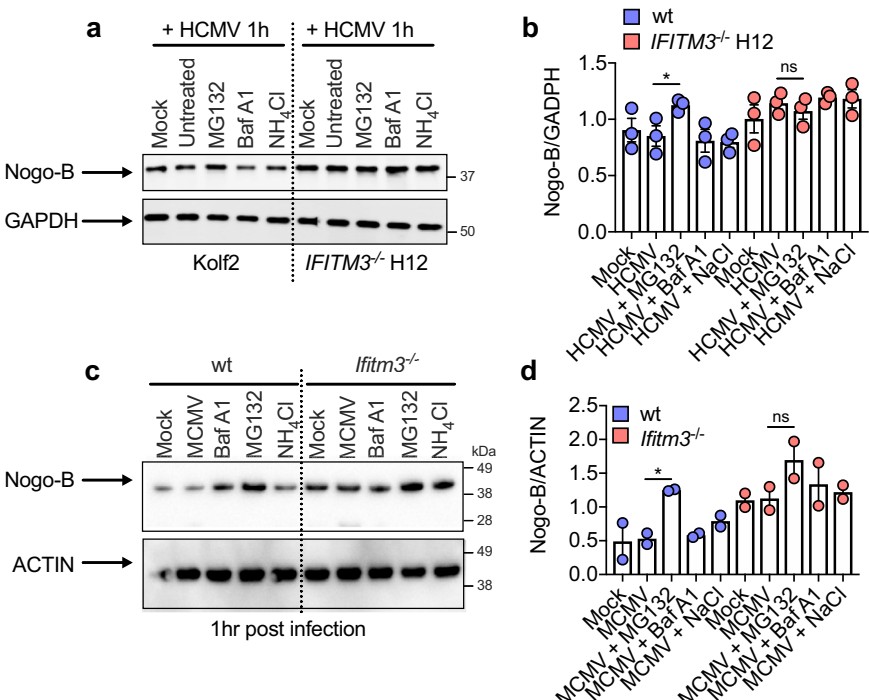

**Fig. 6 | IFITM3 regulates Nogo-B expression via the ubiquitin-proteasome pathway. a**, **b** Kolf2, *IFITM3*⁻/⁻ iPS-DCs or **c**, **d** BM-DCs (from two male donor mice/group) were pre-treated with Bafilomycin A1, NH₄Cl or MG132 for 1 h, then stimulated with (**a**, **b**) HCMV (iPS-DCs) (MOI 5) for 1 h or infected with (**c**, **d**) MCMV (MOI 1) for 1 h (BM-DCs). Nogo-B levels were assayed by Western blot (**a** iPS-DCs, **c** BM-DCs), with results from three independent replicates quantified relative to GAPDH (**b** iPS-DCs) or relative to ACTIN (**d** BM-DCs). Data are shown as mean ± SEM (**b**, **d**). Statistical significance was assessed using Student's *t* test for relevant comparisons. *p* values are reported as follows: n.s., >0.05 and *, ≤0.05. Source data are provided as a Source Data file.

rather than trafficking per se. Interestingly, Nogo-B regulates the generation of sphingolipids, which are key components of lipid rafts[55,56]. Nogo-B is also enriched in lipid domains[57] which may also directly impact on TLR activation. Thus, Nogo-B may alter TLR signalling dynamics by influencing lipid raft structure which may be relevant for TLRs located either at the cell surface or endosomal compartment.

While reticulon proteins have been implicated in peripheral ER shaping[58], the data here support an IFITM3-regulated interaction with TLRs at the plasma membrane and in endosomal compartments. Other non-ER examples of plasma membrane and endosomal partners for Rtn proteins include BACE proteinases[59], Cis-prenyltransferases[60,61], LILRB2[62,63] and Rtn4R[64,65]. Thus, the regulation of TLR function is consistent with other non-ER Rtn roles. Our IP studies revealed the presence of both Rtn3 and Nogo/Rtn4 in IFITM3 complexes. While functional studies of Nogo/Rtn4 deletion yielded prominent effects on CMV-induced cytokine production, the role of Rtn3 may be partially redundant and combined Rtn3/4 deletion may yield even greater effect. While the current study focuses on viral infection and dendritic cells, the prominent expression of Rtns in the brain suggests that the mechanisms defined here may contribute in the regulation of TLR function for a range of neuro-inflammatory diseases.

The exact mechanisms through which IFITM3 promotes proteasome-dependent Nogo-B turnover and how exactly this Nogo-B-Ifitm3 axis influences TLR signalling requires further detailed investigation. However, it is clear from our studies that in IFITM3-deficient cells where we observed increased Nogo-B expression, we also observed heightened inflammatory cytokine production and internalisation of TLR2. It was initially thought that TLRs were partitioned into surface or endosomal TLRs, with all signalling occurring in either location. However more recent studies have demonstrated activation of different inflammatory mediators, dependent on differential TLR

locality within the cell[66,67]. Interestingly, internalisation of cell-surface TLR2 into endosomal compartments in human monocytes is required for NFκB activation[49], and TLR2-induced IL-6 expression can be restricted by endocytosis inhibition[68]. Our observation that inhibition of endocytosis or NFκB induces a similar reduction of IL-6 production only in IFITM3-deficient cells implies that a similar process may be suppressed in IFITM3-expressing DCs to reduce virus-induced inflammation.

Interestingly, the work of Kimura et al. that revealed a role for Nogo-B in regulation of TLR locality in macrophages reported no impact of Nogo-B on TLR2-induced cytokine production[44], in contrast to our data. These differences may reflect the different cell types studied. Indeed, signalling pathways downstream of TLR2 have previously been observed to be divergent in DCs and macrophages[69], implying differential regulation of TLR-induced cytokine responses in different myeloid cells. In support of this hypothesis, Ifitm3 has no impact on MCMV-induced cytokine responses by murine macrophages[23].

In our study, we used iPS-DCs as a model for human myeloid cells. These cells provide a useful model for studying human cells in vitro due to their unlimited numbers and the ability to genetically edit the iPSC progenitors. In these cells, IFITM3 did not directly restrict IAV entry, in contrast to other cell types[70]. It has been shown that CD141⁺ human DCs are resistant to productive infection by IAV[71]. Here, we measured viral NP expression, which is present in viral particles, suggesting that uptake of virus is not affected by IFITM3. Similarly, the relatively low MOI for HCMV used in our studies also led to no productive infection despite significant virus-induced cytokine production. Also, IFITM3 suppressed SARS-CoV-2 induced IL-6 production despite no known role for IFITM3 in restricting SARS-CoV-2 cell entry. Therefore, in these human experimental systems we were able to disentangle the immune-regulatory and antiviral functions of IFITM3 in response to evolutionary diverse viruses.

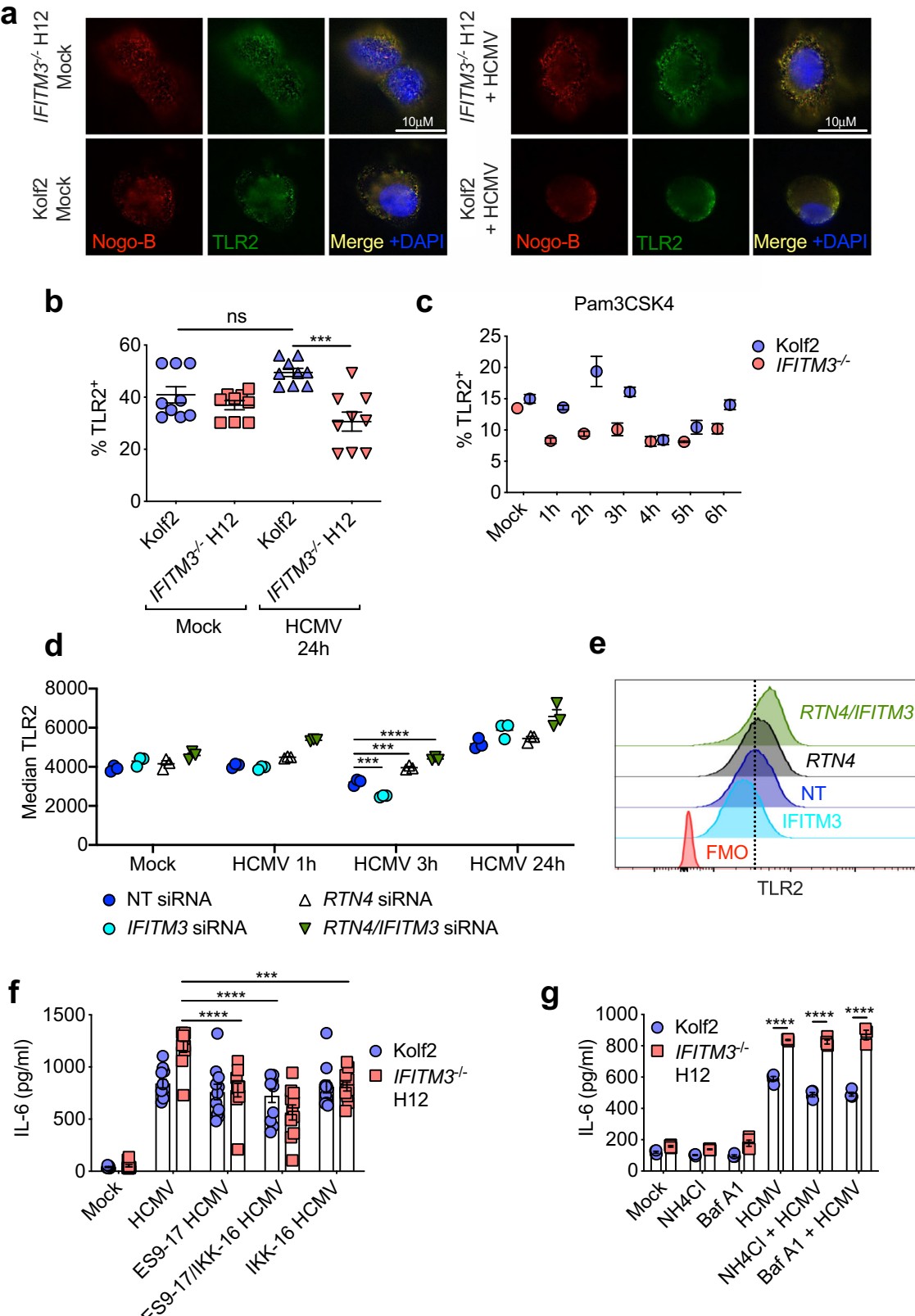

For many viral diseases, effective treatments remain elusive, and vaccines are not readily available or quickly distributed in a pandemic scenario. Therefore, immunomodulatory approaches to reduce host inflammation during viral infection can provide attractive alternatives for rapid treatment of infected individuals, as highlighted by the SARS-CoV-2 pandemic[72]. Our study reveals Nogo-B as an important interacting binding partner of IFITM3 and highlights the previously unappreciated role for both IFITM3 and Nogo-B in influencing virus-induced inflammatory events. Targeting this interaction and dissecting the downstream impacts of this on innate immune recognition may help to expand strategies for controlling pathologic viral induced inflammation.

**Fig. 7 | TLR dynamics are altered in IFITM3-deficient DCs. a** Immunostaining for anti-Nogo-B (Red) and anti-TLR2 (Green) in iPS-DCs mock treated or stimulated with HCMV (MOI 5), (Blue; DAPI). **b** Surface TLR2 was assessed by flow cytometry in Kolf2 and *IFITM3*$^{-/-}$ iPS-DCs either mock treated or stimulated for 24 h with HCMV. Data are shown as mean values from biological separate cell cultures ($n = 9$) ± SEM. **c** iPS-DCs ($n = 3$ separate cell cultures in different wells) were stimulated for 0–6 h with TLR2 ligand Pam3CSK4 and % TLR2 was quantified. Data are represented as mean ± SEM. **d, e** Surface TLR2 was assessed by flow cytometry in HCMV-stimulated THP-1s treated for 72 h with *RTN4/Nogo*, Non-targeting (NT) and/or *IFITM3* siRNAs. Individual cell cultures ($n = 3$) + mean (**d**) and representative histograms (**e**) are shown. Data are expressed as mean MFI ± SEM (**d**). **f, g** IL-6 production by healthy control Kolf2 or *IFITM3*$^{-/-}$ H12 iPS-DCs pre-treated for 1 h with (**f**) endocytosis inhibitor (ES9-17) and/or NFκB inhibitor (IKK-16), or (**g**) Bafilomycin A1 and NH$_4$Cl, followed by stimulation with HCMV (MOI 5) for 24 h. Data presented are from at least two independent experiments, with samples run in at least triplicate for each ELISA. Mean ± SEM are shown from 11 (**f**) or 4 (**g**) biologically independent cell cultures. Statistical significance was assessed using one-way ANOVA with Tukey's multiple comparisons test (**b, d**) or Dunnett's repeated measures test comparing experimental groups with HCMV-alone group in either Kolf2 or *IFITM3* H12 cells (**f**). **g** Statistical significance was assessed using one-way ANOVA analysis with Tukey's multiple comparisons test. *p* values are reported as follows: n.s., >0.05; *, ≤0.05; **, ≤0.01; ***, ≤0.001; and ****, ≤0.0001. Source data are provided as a Source Data file.

## Methods

### Mice, viral infections and treatments
All in vivo experiments were performed according to UK Home Office guidelines and were performed at Cardiff University (Reference: P7867DADD) in specific pathogen-free/SPF conditions at ambient temperature and humidity. *Nogo-A/B*$^{-/-}$ mice[73], and IFITM3-deficient (*Ifitm3*$^{-/-}$) and wt control mice have been described previously[1] and were crossed with *Myd88*$^{-/-}$ (Jackson Laboratory) or for some experiments *Nogo-A/B*$^{-/-}$ mice to generate *Ifitm3*$^{-/-}$/*Myd88*$^{-/-}$ and *Ifitm3*$^{-/-}$/*Nogo-A/B*$^{-/-}$ mice. Novel strains are available upon request. Age- and sex-matched mice between 7 and 12 weeks of age were used in the experiments. *Tlr3*$^{-/-}$[74], *Tlr7*$^{-/-}$[75] and *Tlr9*$^{-/-}$[76] mice were a kind gift from Caetano Reis e Sousa (Crick Institute, London). Relevant wild type control mice were all bred inhouse. MCMV (pSM3fr-MCK-2fl BACmid) was grown and titred using 3T3 cells (ATCC, CRL-1658) with a carboxycellulose overlay. Mice were infected via intraperitoneal (i.p.) injection with between $5 \times 10^5$ to $2 \times 10^6$ PFU MCMV. For Anakinra treatment, mice were injected i.p. with Anakinra (KINERET: Cardiff & Vale NHS Pharmacy) (25 mg/kg) or PBS control on day 0 p.i. For infectious virus quantification form harvested tissue, viral load was determined via plaque assay as previously described[77]. All mice were euthanized using carbon dioxide.

### Antibodies
All antibodies used in experiments are described in Supplementary Tables 1–5.

### HCMV preparation for human in vitro assays
HCMV strain Merlin (with mutations in RL13 and UL128 to enable stable propagation in fibroblasts) or TB40-BAC4 were propagated in immortalised human foetal foreskin fibroblasts (HFFF-hTERT) which were grown in DMEM (Gibco; Invitrogen) containing 10% FCS (Gibco; Invitrogen) at 37 °C in 5% CO$_2$. Cells were pelleted by low-speed centrifugation, then virus pelleted from the supernatant by centrifugation ($29,000 \times g$, 2 h, 21 °C), before being titred by plaque assay over 14 days on HFFFs using 1% Avicel overlay. Cultures that exhibited small plaques at 14 days were re-overlayed and incubated for a further 7 days then recounted. Plaques were visualised using a Zeiss Axio Observer Z1 microscope. Sizes were computed using ImageJ[78,79].

### SARS-CoV-2 propagation
Viral strain used: Victoria 01/20 (BVIC01)[80] (provided by PHE Porton Down after supply from the Doherty Centre Melbourne, Australia). Virus was propagated on VeroE6 cells (ATCC, CRL-1586) and virions were pelleted through a 30% sucrose cushion and titrated on VeroE6 cells[81].

### Generation of z-DC/*Ifitm3*$^{-/-}$ chimeric mice
Female wt recipient mice were gamma irradiated with 550 rad for 2 min over two intervals. After 24 h, mice were intravenously (i.v) transfused with $1 \times 10^6$ bone marrow-derived cells from *Ifitm3*$^{-/-}$ and wt-zDC-DTR[82] female donors. Twelve weeks later mice were administered either with or without Diptheria toxin (DT) (1 µg/ml) i.p. 24 h later mice were infected with MCMV and weight loss was assessed over time.

### In vitro infections
Bone marrow-derived dendritic cells (BM-DCs) cells from either wt, *ifitm3*$^{-/-}$, *Tlr3*$^{-/-}$, *Tlr7*$^{-/-}$ and *Tlr9*$^{-/-}$, *Nogo-A/B*$^{-/-}$, *Nogo-A/B*$^{wt}$, *Ifitm3*$^{wt}$-*MyD88*$^{wt}$, *Ifitm3*$^{-/-}$*MyD88*$^{wt}$, *Ifitm3*$^{wt}$*MyD88*$^{-/-}$, *Ifitm3*$^{-/-}$*MyD88*$^{-/-}$ and *Ifitm3*$^{-/-}$*Nogo-A/B*$^{-/-}$ mice were incubated at $4 \times 10^5$ cells per ml in R10 media supplemented with 100 U/ml Penicillin/streptomycin, 2 mM L-glutamine, 0.1 M HEPES, 50 µM B-Mercaptoethanol and $1 \times$ MEM non-essential amino acids (all Gibco, Thermo Fisher) and 20 ng/ml GM-CSF (Biolegend) for 9 days replenishing the media at d2, 4 and 7. Differentiated BM-DCs were then stimulated with MCMV at a MOI of 1 or 0.1 or IAV A/X-31 (H3N2) at an MOI of 1 as indicated. Cells were infected via centrifugation and supernatant was harvested at 6 and 24 h post infection. In some experiments, cells were treated with various blocking reagents, TLR ligands or siRNAs as described in more detail below.

### iPSC line generation and culture
The healthy control human iPSC line Kolf2 was acquired through the Human Induced Pluripotent Stem Cells Initiative Consortium (HipSci; www.hipsci.org), through which it was also characterised[83]. Consent was obtained for the use of cell lines for the HipSci project from healthy volunteers. Prior to differentiation, iPSCs were grown feeder-free using the Essential 8 Flex Medium kit (Thermo Fisher) on Vitronectin (VTN-N, Thermo Fisher) coated plates as per manufacturer's instructions to 70–80% confluency. iPSCs were harvested for differentiation using Versene solution (Thermo Fisher).

### Generation of *IFITM3*$^{-/-}$ iPSCs
The Wellcome Trust Sanger Institute core gene-editing pipeline generated *IFITM3*$^{-/-}$ iPSC lines, as previously described in ref. 31. The knockout of IFITM3_F01 was generated by a single T base insertion in the first exon using CRISPR/Cas9 in the Kolf2_C1 human iPSC line (a clonal derivative of Kolf2). This was achieved by nucleofection of $10^6$ cells with Cas9-crRNA-tracrRNA ribonucleoprotein (RNP) complexes. Synthetic RNA oligonucleotides (Supplementary Table 6, WGE CRISPR ID: 1077000641, 225pmol crRNA/tracrRNA) were annealed by heating to 95 °C for 2 min in duplex buffer (IDT) and cooling slowly, followed by addition of 122 pmol recombinant eSpCas9_1.1 protein (in 10 mM Tris-HCl, pH 7.4, 300 mM NaCl, 0.1 mM EDTA, 1 mM DTT). Complexes were incubated at room temperature for 20 min before electroporation. After recovery, cells were plated at single cell density and colonies were picked into 96-well plates. In total, 96 clones were screened for heterozygous and homozygous mutations by high throughput sequencing of amplicons spanning the target site using an Illumina MiSeq instrument (for primer sequences, see Supplementary Table 6). Final cell lines were further validated by Illumina MiSeq (for sequence information, see Supplementary Table 7). Two homozygous targeted clones were used in downstream differentiation assays.

## Differentiation of iPSCs to iPS-DCs

Differentiation of iPSCs to dendritic cells and macrophages. To differentiate iPSCs to dendritic cells, slight modifications were made to a previously published protocol[84]. Briefly, upon reaching confluence, iPSCs were harvested and plated into Essential 8 Flex medium supplemented with 50 ng/ml bone morphogenetic protein 4 (BMP-4; Bio-Techne), 20 ng/ml stem cell factor (SCF; Bio-Techne), 50 ng/ml vascular endothelial growth factor (VEGF; Peprotech EC Ltd.), and 50 ng/ml GM-CSF (Peprotech EC Ltd.) in ultralow attachment (ULA) plates (Corning). The medium was changed to X-VIVO-15 (Lonza), with sequential removal of BMP-4 by day 5, VEGF by approximately day 14, and SCF by approximately day 19. In addition, IL-4 (Peprotech EC Ltd.) was added sequentially in increasing concentrations, starting from approximately day 12 at 25 ng/ml and increasing to 100 ng/ml by approximately day 20. By day 20, floating immature DCs were harvested from ULA plates, filtered through 70-μm filters (Corning), counted, and seeded at $1 \times 10^6$ per well of 6-well CellBind plates (Corning) in X-VIVO-15 medium supplemented with 100 ng/ml IL-4 and 50 ng/ml GM-CSF. iPS-DCs were used for assays at the immature phase between 4 and 5 days post-seeding in CellBind plates. For the assays, floating iPS-DCs were harvested from differentiation plates, washed with PBS, counted, and seeded in X-VIVO-15 medium without cytokines at an assay-dependent concentration.

## Genotyping of human donors

The use of human material was approved by Oxford Tropical Research Ethics Committee (OxTREC) under code OXTREC 1001-13. All volunteers were provided with full details of the project and provided written informed consent and were not compensated. *IFITM3* rs12252 genotype was identified for each participant through PCR amplification of the *IFITM3* gene (primers, Supplementary Table 6). In total, 100 ng of genomic DNA was added to Taq DNA polymerase (PCR Biosystems) and samples were run at: 95 °C 15 s, 58 °C 15 s, 72 °C 15 s for 35 cycles. Amplification was confirmed by running samples on a 2% agarose gel prior to PCR Clean up (QiaQuick PCR Purification kit, Qiagen). Samples were sequenced using Sanger sequencing (Source Bioscience) and rs12252 was identified from the DNA Electropherogram File (.ab1 file).

## Generation of blood-derived human dendritic cells

PBMCs from three independent donors were isolated from leukapheresis products using Lymphoprep density gradient centrifugation and SepMate PBMC isolation tubes (StemCell Technologies), under the Weatherall Institute of Molecular Medicine, University of Oxford Human Tissue Authority license 12433. Human CD14 microbeads were used in combination with LS columns (both Miltenyi Biotec) to positively select CD14$^+$ blood monocytes. CD14$^+$ cells were seeded at a density of $3 \times 10^6$ to $5 \times 10^6$ isolated monocytes in 3 ml of RPMI medium supplemented with 10% heat-inactivated foetal bovine serum (FBS; Sigma-Aldrich), 250 IU/ml IL-4, and 800 IU/ml GM-CSF (both Peprotech EC Ltd) into a 6-well plate and incubated at 37 °C for 2 days. After 2 days, 1.5 ml of medium was removed from each well, and 1.5 ml of fresh medium supplemented with 500 IU/ml IL-4 and 1600 IU/ml GM-CSF was added. After a further 3-day incubation, cells were harvested at the immature phenotype and assayed.

## Stimulation of human DCs with IAV, SARS-COV-2 and HCMV

iPS-DCs, or human mDCs were stimulated with A/X-31 influenza virus, gamma-irradiated A/X-31 influenza virus or SARS-COV-2 at an MOI of 1, or HCMV strain merlin at MOIs stated in each legend, by the addition of virus to a small volume of X-VIVO-15 cell culture supernatant (50 ul for assays in 96-well plates; 200 ul for assays in 24 well plates) followed by incubation at 37 °C for 1 h, after which fresh culture medium was added. After 6 and 24 h inoculation respectively, 150 ul supernatant

was collected and stored at −80 °C. Each infection condition was repeated in triplicate.

## Inhibitor assays

For proteasomal/lysosomal inhibition assays, $5 \times 10^5$ cells per well iPS-DCs (human) or $1 \times 10^5$ BM-DCs (mouse) per condition were pre-incubated for 1 h prior to viral stimulation or infection with inhibitors MG132 (10 μM; Merck Millipore) BafA1 (0.5 μM; Invivogen) or $NH_4Cl$ (10 mM; Sigma-Aldrich), before addition of HCMV for 1 and 3 h or with MCMV for 6 and 24 h as described above, after which DCs were harvested for protein preparation or supernatants harvested. Further inhibition assays were performed using $1 \times 10^5$ cells per well BM-DCs (mouse) per condition. In some experiments, BM-DCs were pre-incubated for 1 h prior to viral infection with TLR7 synthetic peptide (2 μg/ml; Thermo Fisher), ODN 2088 (10 μM; Invivogen), 3-Methyladenine (3MA, 5 mM; Merck Millipore) or Anakinra (500 ng/ml; Cardiff & Vale NHS Pharmacy). Protein lysates were then generated from cells and/or culture supernatants were harvested for cytokine analysis. For human blocking assays that target endocytosis and NFκB signalling, ES9-17 (endocytosis inhibitor; 100 μM; Merck Millipore) and/or IKK-16 (NFκB inhibitor; IKK inhibitor VII, 1 nM; Cambridge Bioscience) were added 1 h prior to addition of HCMV. Supernatants were harvested after 24 h. For assays using neutralising antibodies to TLR2 (200 μg/ml; Invivogen) or HCMV (500 μg/ml; Cytotect CP Biotest) antibodies were added 1 h prior to addition of HCMV or TLR2 ligand.

## Preparation of RNA and RT-qPCR

iPS-DCs were harvested from the plates, and RNA was prepared using the RNeasy Mini kit (Qiagen). RNA was reverse transcribed with the QuantiTect reverse transcription (RT) kit (Qiagen), according to the manufacturer's protocol. All RT-qPCR experiments were performed with TaqMan gene expression assays and TaqMan gene expression master mix (Applied Biosystems) on the Applied Biosystems StepOne real-time PCR system. IFITM1 (Hs00705137_s1), IFITM2 (Hs00829485_sH) RT-qPCR data were analysed via the comparative threshold cycle ($C_T$) method with glyceraldehyde 3-phosphate dehydrogenase (GAPDH) (Hs02758991_g1) as an endogenous control (Thermo Fisher).

## TLR ligand stimulation

Human iPS-DCs were plated at $2 \times 10^4$ cells per well in 200 μl of X-VIVO-15 medium without cytokines. TLR ligands were added directly to the medium, and supernatants were harvested after a 24 h incubation at 37 °C. For the assays, TLR ligands were used at the following concentrations: TLR2; Pam3CSK4, 300 ng/ml (InvivoGen); TLR3; Poly(I:C), 50 μg/ml (InvivoGen); TLR4; Lipopolysaccharide (LPS), 500 ng/ml (Sigma-Aldrich); TLR7; Imiquimod, 50 μg/ml (InvivoGen); and TLR9; ODN 2216, 3 μg/ml (Miltenyi Biotech). Mouse BM-DCs were plated out at $1 \times 10^5$ cells per well. TLR ligands were added directly to the well as before with supernatants harvested at 6 and 24 h at 37 °C. TLR ligands were used at the following concentrations: TLR2; Pam3CSK4, 0.5 μg/ml (InvivoGen); TLR3; Poly(I:C), 10 μg/ml (InvivoGen); TLR4; LPS, 10 μg/ml (InvivoGen); TLR5; Flagellin 5 μg/ml (InvivoGen); TLR7; Imiquimod, 5 μg/ml (InvivoGen); and TLR9; CpG Class B ODN 1826, 0.05 μM (Invivogen).

## Cytokine analysis

Human and mouse IL-6 and TNF protein were measured by an enzyme-linked immunosorbent assay (ELISA) (BioLegend) according to the manufacturer's instructions.

## Proteomic pulldowns

BM-DCs cells from either wt or *Ifitm3*$^{-/-}$ mice were grown in SILAC RPMI media (Gibco, Thermo Fisher) either supplemented with 10% HI

dialysed and filtered FCS (Sigma-Aldrich) 0.1 M HEPES, 50 μM B-Mercaptoethanol (both Gibco, Thermo Fisher), and either L-Lysine-2HCl 13C6 15N2 and L-Arginine-HCl 13C6 15N4 ('Heavy' amino acids) or L-Lysine-2HCl 4,4,5,5-D4 and L-Arginine-HCl 13C6 ('Medium' amino acids) (all Cambridge Isotope Laboratories). Wt cells were grown in 'Medium' SILAC media and *Ifitm3*−/− were grown in 'Heavy' SILAC media. BM-DC cells were differentiated as described above for 10 days and were infected or not with MCMV at an MOI of 1 for 3 h. Cells were removed from the plates post infection and subsequently lysed using Pierce™ IP lysis buffer (Thermo Fisher) supplemented with 1 M proteasome inhibitors (Sigma-Aldrich). IP for IFITM3 (α-fragillis, 1 μg/ml; Abcam) and anti-Nogo-B (1 μg/ml, R&D Systems) was performed on all samples as described previously using Pierce™ Protein Plus Agarose A/ G beads (Thermo Fisher)[85]. To confirm specificity of IP, control anti-rabbit IgG (1 μg/ml; Abcam) was also performed. Post IP beads were that were bound to IFITM3 only were combined and eluted from the Agarose using 1 × NuPAGE™ LDS sample buffer (Thermo Fisher) supplemented with 100 mM DTT (Sigma-Aldrich). Samples were run on a NuPAGE™ 4 to 12% Bis/Tris gels (Thermo Fisher) running ~0.5–1 cm into the gel. The gel was stained using Colloidal blue staining kit (Thermo Fisher) as per manufacturers recommendations. The stained lane was excised and cut into six fragments. Following in-gel reduction and alkylation, proteins were digested using trypsin, and the resulting peptides were eluted and dried prior to analysis on an Orbitrap Lumos mass spectrometer (Thermo Fisher). Loading solvent was 3% MeCN, 0.1% FA, analytical solvent A: 0.1% FA and B: MeCN + 0.1% FA. All separations were carried out at 55 °C. Samples were loaded at 5 μl/min for 5 min in loading solvent before beginning the analytical gradient. The following gradient was used: 3–40% B over 29 min followed by a 3 min wash at 95% B and equilibration at 3% B for 10 min. The following settings were used: MS1: 300-1500 Th, 120,000 resolution, $4 \times 10^5$ AGC target, 50 ms maximum injection time. MS2: Quadrupole isolation at an isolation width of *m/z* 1.6, HCD fragmentation (NCE 35) with fragment ions scanning in the Orbitrap from *m/z* 110, $5 \times 10^4$ AGC target, 60 ms maximum injection time, ions accumulated for all parallelisable time. Dynamic exclusion was set to ±10 ppm for 60 s. MS2 fragmentation was trigged on precursors $5 \times 10^4$ counts and above. Data were analysed in MaxQuant version 2.0.1.0, and the method of Significance A was used to estimate *p* values[86]. Full dataset is provided in the Supplementary files.

## Protein preparation and western blotting
IFITM3 western blots using human cells were performed as described in refs. 16, 31. For other proteins, $1 \times 10^6$ cells per condition/cell line were homogenised using RIPA lysis buffer (Thermo Fisher). Protein extracts were prepared for gel electrophoresis by addition of 1:1 Novex™ Tris-Glycine SDS Sample Buffer (Thermo Fisher) and heating at 85 °C for 5 min, followed by addition of sample to 12% or 8–16% (dependent on predicted band size) Novex wedgewell Tris Glycine mini gels (Thermo Fisher) for electrophoresis. Proteins were blotted onto PDVF membrane using the mini-gel wet-transfer XCell II Blot Module (Thermo Fisher) in transfer buffer (20% methanol, 25 mM Tris-base and 192 mM Glycine). Membranes were blocked with 5% milk powder in TBS-T (Sigma-Aldrich), then transferred to primary antibody for anti-Nogo-B (0.2 μg/ml; Bio-Techne), anti-GAPDH (1 μg/ml; Merck Millipore), anti-IFITM1 (3 μg/ml; Proteintech), anti-IFITM2 (3 μg/ml; Proteintech) or anti-Nogo-A (1 μg/ml; Abcam) in 5% milk/TBS-T and incubated overnight at 4 °C. Primary antibodies were probed with IRDye 680LT goat anti-mouse (Li-Cor, 926-68020), IRDye 800LT goat anti-rabbit (Li-Cor, 925-32210) or Alexa Fluor 680 donkey anti-sheep IgG (H + L; Invitrogen), and visualised using the Li-Cor Odyssey Imaging System. Protein band intensity was determined using ImageJ, with values calculated relative to GAPDH with background removed for the same sample lane. In murine cells, $1 \times 10^5$ cells per condition/cell line were homogenised using 1 × NuPAGE™ LDS sample buffer (Thermo

Fisher) supplemented with 100 mM dithiothreitol (DTT) (Sigma-Aldrich) and boiled at 100 °C for 10 min. Samples were run using a NuPAGE™ 4 to 12% Bis-Tris gels (Thermo Fisher) for electrophoresis. Proteins were blotted onto PDVF membrane using the mini-gel wet-transfer XCell II Blot Module (Thermo Fisher) in transfer buffer (1 × NuPAGE™ Transfer Buffer, Thermo Fisher Scientific). Membranes were blocked with 5% milk powder in PBS-T (Sigma-Aldrich), then transferred to primary antibody for anti-Nogo-B (0.2 μg/ml; Bio-Techne), anti-fragilis (IFITM3, 1 μg/ml; Abcam) and anti-ACTIN (1 μg/ml; Sigma-Aldrich) in 0.5% milk/PBS-T and incubated overnight at 4 °C. Primary antibodies were probed with either rabbit-anti-sheep IgG (H + L)-HRP conjugate, goat anti-rabbit IgG (H + L)-HRP conjugate (both Biorad) or Veriblot IP detection reagent (Abcam) and visualised using a Syngene G:Box imaging system. Protein band intensity was determined using ImageJ, with values calculated relative to ACTIN with background removed for the same sample lane. All uncropped western blots are provided within the Source Data file.

## Immunostaining and imaging
iPS-DCs were harvested from plates, washed in PBS, fixed in 4% PFA for 10 min at room temperature, and then spun onto 0.01% poly-L-lysine (Sigma-Aldrich) coated slides using a Cytospin cytocentrifuge. Immediately after spinning onto slides, residual aldehydes were quenched using 25 mM glycine for 10 min, followed by three washes with PBS. Cells were permeabilised with 0.5% saponin (Fisher Scientific), followed by blocking with 0.05% saponin and 2% BSA (Sigma-Aldrich) for 1 h, and incubation with primary antibodies for anti-TLR2 (Abcam) and anti-Nogo-B (Bio-Techne) diluted 1:100 in blocking solution for 1 h. Samples were washed with 0.05% saponin three times, and then fluorescently conjugated secondary antibodies (Donkey anti-sheep IgG Cy5, Merck Millipore; Rabbit anti-goat IgG FITC, Abcam) diluted in blocking solution were added for 30 min. Samples were washed three times with 0.05% saponin and then nuclei were counterstained with DAPI (NucBlue; Invitrogen). Samples were washed three times and then mounted with Vectashield (Vector Laboratories) and left to dry overnight before imaging on a DeltaVision Elite system, with images captured using a CoolSNAP HQ2 camera. For colocalization analysis, z-stacked images were deconvolved with Huygens deconvolution software (SVI), followed by quantification of colocalization and statistical summarisation using Manders overlap coefficient on ImageJ.

For subcellular co-localisation studies, THP1 cells (ATCC, TIB-202) were infected/not with MOI 5 of pentamer-deficient Merlin HCMV for 3 h. Cells were fixed and permeabilized by ice cold acetone fixation for 10 min and spun onto slides as described above. Cells were then stained with antibodies (anti-RAB7, Biolegend, clone W16034A; anti-IFITM3[31]; anti-Nogo B, R and D systems, anti-CD107a anti-LAMP1, Biolegend, clone H4A3) and autofluorescence was blocked using True-VIEW Autofluorescence Quenching Kit SP-8400, nuclei were stained with Hoechst 33342 Invitrogen H3570 and slides were mounted with VECTASHIELD[R] Vibrance™ Antifade Mounting Media H1700-2. Images were captured using a Zeiss LSM 800 confocal microscope and Zen blue 2.6 software and images produced using ImageJ.

## THP-1 culture and siRNA
The human monocytic cell line THP-1 was used for siRNA knockdown experiments. THP-1 cells were cultured in RPMI 1640 (Gibco, Thermo Fisher) supplemented with 10% FBS, 100 U/ml Penicillin/streptomycin and 2 mM L-glutamine (Gibco, Thermo Fisher). For siRNA knockdown, THP-1s were transferred to Accell siRNA Delivery medium (Horizon Discovery). Cells were targeted with 1 μM non-targeting siRNA and/or SMARTPool (Horizon Discovery) directed against *RTN4/Nogo* (Catalogue number: E-010721-00-0050), *IFITM3* (Catalogue number: E-014116-00-0020) or control (Catalogue number: A-004253-14-0010). In total, 72 h post-targeting, THP-1s were plated for assays at $1 \times 10^5$ cells per condition per replicate. Knockdown was assessed at 72 h post-

targeting by western blot for IFITM3 and Nogo-B expression, as described above. For murine BM-DC siRNA transfection, cells were differentiated as described above. During differentiation (d7), $1 \times 10^6$ BM-DCs were transfected with 150 nM siRNA targeting either *Rtn4/Nogo* (Catalogue number: 1027417 SI01408512, Qiagen) or control siRNA (Mus musculus flexitube siRNA (*Rtn4/Nogo*) or AllStars negative control siRNA (control)) (Catalogue number: 1027280, Qiagen), using HiPerfect transfection reagent (Qiagen). Cells were left for 24 h and then a repeat transfection with siRNAs was performed as before (d8 of differentiation). Cells were used at d9 of differentiation using $1 \times 10^5$ cells per condition per replicate. Knockdown was assessed at 48 h post initial transfection by western blot for Nogo-B expression, as described above.

### Flow cytometry

Flow cytometry was performed as described and using gating strategies in ref. 32. Briefly, for analysis of surface markers on iPS-DCs or Monocyte-derived DCs, cells were stained with Zombie Aqua fixable dye (BioLegend), Fc receptors were blocked using human TruStain FcX (BioLegend), and cells were then subsequently stained for iPS-DC surface markers anti-CD11c-FITC (Bu15, Biolegend) and anti-CD141-APC (M80, Biolegend) or Monocyte-derived DCs surface markers anti-HLA-DR (L243, Biolegend) and anti-CD209 (DC-SIGN) (9E9A8, Biolegend). For the detection of TLR2, cells were also stained anti-TLR2-PE (TL2.1, Biolegend) or TLR2-APC (W1514C, Biolegend) prior to fixation with 4% paraformaldehyde. Data were acquired using an Attune NxT flow cytometer (Thermo Fisher). Electronic compensation was performed with antibody (Ab) capture beads (BD Biosciences) stained separately with individual monoclonal antibodies used in the experimental panel. Data were analysed using the FlowJo software (TreeStar, Inc.).

### Statistical analyses

Statistical significance was performed using the GraphPad Prism software. Mass spectrometry statistical *p* values were estimated using Benjamini–Hochberg-corrected significance A values[86]. A *p* value ≤0.05 was considered to be significant. For all other tests performed, details are provided in the figure legends. *p* values are reported as follows: n.s., >0.05; *, ≤0.05; **, ≤0.01; ***, ≤0.001; and ****, ≤0.0001.

### Reporting summary

Further information on research design is available in the Nature Research Reporting Summary linked to this article.

## Data availability

The mass spectrometry proteomics data have been deposited to the ProteomeXchange Consortium via the PRIDE partner repository under the dataset identifier PXD035254. All data are included in the Supplementary Information or available from the authors upon reasonable requests, as are unique reagents used in this article. Source data are provided with this paper.

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

## Acknowledgements

This work was funded by a Wellcome Trust Senior Research Fellowship to I.R.H. (grant 207503/Z/17/Z); a Wellcome Trust Senior Research Fellowship to M.P.W. (grant 108070/Z/15/Z); the Medical Research Council, United Kingdom (grant MR/L018942/1 and MRC Human Immunology Unit Core); and the Chinese Academy of Medical Sciences (CAMS) Innovation Fund for Medical Sciences (CIFMS), China (grant 2018-I2M-2-002). This work was also partly funded by the MRC/NIHR through the UK Coronavirus Immunology Consortium (CiC; MR/V028448/1). P.S. was funded by a Cardiff University Systems Immunity University Research Institute PhD Studentship. The Wellcome Trust Sanger Institute was the source of the Kolf2 human induced pluripotent cell line, which was generated under the Human Induced Pluripotent Stem Cell Initiative funded by a grant from the Wellcome Trust and the Medical Research Council, supported by the Wellcome Trust (grant WT098051) and the NIHR/Wellcome Trust Clinical Research Facility. IFITM3 iPSC knockout lines were generated, characterised and banked by the Gene editing core facility at the Wellcome Sanger Institute. The authors would like to thank Peter Wing (NDM, University of Oxford) for his help with SARS-CoV-2 propagation.

## Author contributions

M.C., J.L.F., M.M., P.S., D.W., S.D., S.C., K.H., L.N., R.A., B.J., M.C., S.M.-N., and J.A.L. performed experiments; M.C., J.L.F., D.W., L.N., R.A., M.P.W., R.J.S., T.D., and I.R.H. analysed the data; S.M.S., T.D., M.P.W., and R.J.S. provided reagents; I.R.H. directed the study; M.C., J.L.F., S.M.S., and I.R.H. wrote the manuscript.

## Competing interests

The authors declare no competing interests.

### Ethical approval

All animal studies were performed at Cardiff University (Heath Park research support facility) under UK Home Office Project License number P7867DADD, as approved by the UK Home Office, London, United Kingdom. Written consent was obtained for the use of cell lines for the HipSci project from healthy volunteers. Ethical approval was granted by the National Research Ethics Service (NRES) Research Ethics Committee Yorkshire and The Humber-Leeds West, under reference number 15/YH/0391.
