## [Peer Review File · Nature Communications]

REVIEWER COMMENTS

Reviewer #1 (Remarks to the Author):

In this study the authors have followed up their previous observations that suggested that IFITM3 negatively regulates the inflammatory response to viral infection through reduction of IL6. Here they show in both human and murine DCs that in the absence of IFITM3, the IL6 response to various viruses via TLRs is increased, and this specifically dependent on Myd88. They go on to screen IFITM3 interaction factors that might promote negative regulation of TLR responses and find RTN4/NOGO-1, an ER resident protein previously implicated in TLR function, as an IFITM3 binding partner. RTN4 knockout/down reverses the enhanced IL6 responses when IFITM3 is deleted, confirming a functional interaction. The authors further show that upon viral stimulation, IFITM3 appears to promote RTN4 degradation. In its absence, RTN4 and TLRs co-localize in endosomes and at the cell surface.

This is a very interesting and important paper that sheds considerable light on the role of IFITM3 in systemic inflammatory responses in viral infection independent of its role in viral entry restriction. It is well written and performed and the results are conclusive. As such this should be of considerable interest to the readers of Nat Comms.

I have very little to criticise about the first 4 figs that make a solid case for a functional interaction between IFITM3 and RTN4 in TLR regulation. My real criticism (if only somewhat minor) is that the cell biology of the mechanism presented in Figures 5 and 6 is less strong than the functional genetic data that precedes it. I appreciate that these are likely to be subtle and subcellular compartment specific. As such, the authors consider strengthening the data in the following way:

- The cell biology of RTN4/TLR localization in Fig6a could be quantified with respect to endosomal markers. It is interesting that IFITM3 is contributing this phenotype, but IFITM2 is not. IFITM2 is only a handful of aa's different from IFITM3 in humans yet localises to different compartments and exerts distinct antiviral activities against enveloped viruses. Does IFITM2 interact with RTN4? And can the authors shed light on where in the cell IFITM3 and RTN4 are interacting?

Reviewer #2 (Remarks to the Author):

This manuscript is a follow up to a previous report from this group showing that the antiviral IFITM3 protein decreases severity of CMV infections, not by directly inhibiting virus entry, but rather by limiting IL-6 production. Here they provide evidence that this mechanism depends on IFITM3 expression in dendritic cells, and further reveal that IFITM3 limits MyD88-dependent CMV pathogenesis using mouse models. They go on to perform Co-IP proteomics on IFITM3 and identify Nogo-B (Rtn4) as a putative interacting partner for IFITM3. Nogo-B has known roles in regulating inflammation, which the authors confirm. They further show that Nogo-B knockout counteracts the hyper-production of IL-6 in IFITM3 knockout cells and mice. Mechanistic studies suggest that IFITM3 regulates levels of Nogo-B and that Nogo-B controls TLR surface expression levels. While these findings are intriguing and exciting, they fall short of fully linking IFITM3 and Nogo-B as direct partners in regulating inflammatory cytokine production during virus infections. Comments and Suggestions aimed at strengthening the manuscript are below.

Major concerns:

1. Does Nogo-B IP also pull down IFITM3?
2. Do IFITM3 and Nogo-B co-localize in cells?
3. The link between IFITM3 and Nogo-B in terms of cytokine regulation is not fully substantiated. In addition to the loss of function studies, can gain of function experiments be performed? For example, what is the effect of IFITM3 overexpression in WT and NogoB KO cells on IL-6 production?
4. How can we be certain that the bands in Fig 5E represent ubiquitinated Nogo-B? What effect

does IFITM3 overexpression have on Nogo-B ubiquitination and levels?

5. Are Nogo-B/IFITM3 double knockout mice available for examination of their in vivo infection phenotype?

Minor comments:

1. Please check the accuracy of this statement in the introduction "...with reduced number of antiviral CD8+ T cells in IAV-infected lung tissue."

2. Everitt, *Nature*, 2012, and Kenney, *PNAS*, 2019 could be added in reference to the following statement in the introduction as IL-6 and other cytokines were upregulated in tissues of IFITM3 KO mice following influenza virus infection in these publications: "Similarly, in mouse models of viral infection, *Ifitm3* deficiency alters cytokine and chemokine profiles and leukocyte influx to sites of infection 19–21"

3. The following statement from the results is incorrect and the reference is cited inappropriately: "Although IFITM3 has a negligible impact on SARS-CoV-2 cell entry 37." The cited publication shows that IFITM3 can have distinct effects on SARS-CoV-2 entry in different cell types. Other publications have also suggested dual functions for IFITM3 in SARS-CoV-2 infections, including Shi, *EMBO J*, 2021, and Winstone, *J Virol*, 2021.

4. Can the authors speculate as to why other known interactors of IFITM3 (e.g., NEDD4, Fyn, AP-2, VAPA, ZMPSTE24) were not pulled down in this study?

5. Relevant comparisons in Fig 5B,D should be graphed next to each other. The current graph layouts are difficult to follow.

We thank the reviewers for their supportive and constructive reviews.

Reviewer #1

We thank the review for commenting that:

'This a very interesting and important paper that sheds considerable light on the role of IFITM3 in systemic inflammatory responses in viral infection independent of its role in viral entry restriction. It is well written and performed and the results conclusive. As such this should be of considerable interest to the readers of Nat Comms'.

I have very little to criticise about the first 4 figs that make a solid case for a functional interaction between IFITM3 and RTN4 in TLR regulation.

We thank the reviewer for their appreciation of this work.

My real criticism (if only somewhat minor) is that the cell biology of the mechanism presented in Figures 5 and 6 is less strong than the functional genetic data that precedes it. I appreciate that these are likely to be subtle and subcellular compartment specific. As such, the authors consider strengthening the data in the following way:

The cell biology of RTN4/TLR localization in Fig6a could be quantified with respect to endosomal markers.

In order to address this question, it would be necessary to stain with TLR2, RTN4 and antibodies against either Rab7 or LAMP1 concurrently. This required either direct conjugates or antibodies from three different species. The former typically gives very weak staining in immune fluorescence, the latter limits significantly the antibodies than can be used.

We attempted multiple combinations to address the question including multiple commercially available TLR2 (clones TL2.1 QA16A01 and TL2.1) and Rab9 (clones EPR13272, D52G8 and Mab9) in addition to the clones used in the data presented below. After no success and limited availability of suitable reagents for Rab9 we looked at Rab7 as an alternative endosome marker and in addition to the W16034A clone used, EPR7589 was also tested. Whilst some limited staining was seen for some of these antibodies when used individually, the staining was lost when used in panels. This issue, together with autofluorescence particularly in infected cells, led to undistinguishable staining above background. Despite various combinations of antibodies and attempts to quench autofluorescence we struggled to achieve reliable staining when antibodies were used in panels for testing in infected cells. Attempts to amplify the staining with tyramides was also unsuccessful.

Whilst some staining of naïve cells was achieved further optimisation is required for triple-staining of infected cells and, we feel, is beyond the scope of this manuscript. Indeed, to investigate this properly, we will need to cross *Ifitm3*^{-/-} or *Ifitm3*^{-/-}*Nogo-B*^{-/-} mice to mice expressing fluorescent tagged TLRs, or use fluorescently-tagged human TLR constructs in *WT/Ifitm3* cells for analysis of multiple subcellular compartments using live cell imaging. This is clearly beyond the scope of the current manuscript.

List and details of the clones tried:

Recombinant Alexa Fluor® 647 Anti-RAB7 antibody [EPR7589] Abcam (ab198337)
CD282 (TLR2) Monoclonal Antibody (clone TL2.1), eBioscience™ 14-9922-82
APC anti-mouse/human CD282 (TLR2) (clone QA16A01) Recombinant Antibody Biologend 153005
PE anti-human CD282 (TLR2) (clone TL2.1) Antibody Biologend 309707.
Recombinant Anti-Rab9 antibody [clone EPR13272] - Late Endosome Marker Abcam(ab179815)
RAB9 Monoclonal Antibody (clone Mab9) Invitrogen MA3-067
Rab9A (Clone D52G8) XP® Rabbit mAb Cell signalling technologies 5118

It is interesting that IFITM3 is contributing this phenotype, but IFITM2 is not. IFITM2 is only a handful if aa's different from IFITM3 in humans yet localises to different compartments and exerts distinct antiviral activities against enveloped viruses. Does IFITM2 interact with RTN4?

We attempted to perform this experiment using *Ifitm3*^{-/-} mice which were readily available controls to assess any cross reactivity of antibodies that are reactive with Ifitm2. However, the only commercially available IFITM2 reagent for mice (Santa Cruz) failed to detect Ifitm2 in input whole cell lysate controls (panel B, below). As expected, it therefore was also unable to detect ifitm2 bound to Nogo-B following immunoprecipitation (Panel A, below). Without an antibody that detected murine Ifitm2, we are therefore unable to perform this experiment. Importantly, however, two separate interactome studies that have validated our RTN4-IFITM3 interactions do not report RTN4-IFITM2 or RTN4-IFITM1 binding. We reference these studies on page 10.

And can the authors shed light on where in the cell IFITM3 and RTN4 are interacting?

As show in Fig. S4B, we detect co-localization of IFITM3 and RTN4 in Rab7⁺ endosomes but not CD107a⁺ lysosomes. We discuss this data on page 11.

Reviewer #2 (Remarks to the Author):

We thank the reviewer for acknowledging that our '*findings are intriguing and exciting*'. We are also very grateful for the reviewer for their constructive criticism which was helped us improve our manuscript.

Major concerns:

1. Does Nogo-B IP also pull down IFITM3?

We attempted to perform this with two different anti-Nogo-B antibody clones (AF6034 and JM02-34). Clone AF6034 was validated to specifically I.P Nogo-B (A). However, when using either clone we did not detect *Ifitm3* binding, and in the case of JM02-34 IFITM3 was pulled down non-specifically during IP from *Nogo-A/B*^{-/-} cells (B). Thus, it was impossible to interpret experiments with JM02-34. For AF6034, it is possible that the antibody binds to an epitope that is shared with the interaction domain for IFITM3; antibodies binding distinct epitopes will need to be generated to address this point. As a result, these experiments could not confirm *Ifitm3*-NogoA/B binding. Importantly, however, as discussed on page 10 the interaction between RTN4 and IFITM3 has been reported in two independent studies using mass spectrometry based approaches of epitope tagged proteins

(Cell, Volume 184, Issue 11, 2021, Pages 3022-3040.e28; Nature Immunology, 2019 Apr;20(4):493-5020). Furthermore, in addition to showing pulldown of NogoB by IP of IFITM3, we now also demonstrate co-localization of IFITM3 and Nogo-B by immune fluorescence (see next point).

2. Do IFITM3 and Nogo-B co-localize in cells?

As shown in Fig. S4A, we detect co-localization of IFITM3 and RTN4 in Rab7⁺ endosomes but not CD107a⁺ lysosomes. We discuss this data on page 11.

3. The link between IFITM3 and Nogo-B in terms of cytokine regulation is not fully substantiated. In addition to the loss of function studies, can gain of function experiments be performed? For example, what is the effect of IFITM3 overexpression in WT and NogoB KO cells on IL-6 production?

We are unclear how the suggested experiment will support the conclusions of our study. The presence of Ifitm3 reduces Nogo-B protein abundance. We have no data suggesting that increasing Ifitm3 in any cellular system will further suppress virus-induced cytokine secretion. Also as highlighted by reviewer 1 when discussing the cell biology of the study, the effects are 'likely to be subtle and subcellular compartment specific'. There is therefore some risk that over-expression approaches will not generate interpretable data.

4. How can we be certain that the bands in Fig 5E represent ubiquitinated Nogo-B? What effect does *IFITM3* overexpression have on Nogo-B ubiquitination and levels?

We are extremely grateful to the reviewer for challenging us on this. To validate ubiquitin expression upon Nogo-B (AF6034) IP in wt and *Ifitm3*^{-/-} BMDCs we first validated that our Nogo-B antibody specifically precipitated Nogo-B (see response to point 1). However, when examining ubiquitin bands following anti-Nogo-B I.P or IgG control using lysis buffer conditions used in our original submission, we were concerned that we saw no meaningful difference between IgG or anti-Nogo-A/B samples (A).

(B) To improve this we used *NogoA/B*^{-/-} cells as controls in combination with increasing salt lysis concentration within the lysis buffer (from 150mM to 500mM) to reduce non-specific binding. We also included another Nogo-B antibody clone JM02-34 in addition to AF6034. However, many non-specific bands were still present throughout the gel after I.P with either antibody, with reduced background in samples precipitated with our original clone (AF6034). (C) Increasing lysis buffer conditions to harsher conditions (2% SDS, 50mM Tris-HCL pH7.5, 250mM NaCl, 1%NP-40, 1mM EDTA, 10mM N-ethylmaleimide and protease cocktail 1/100) used elsewhere (<https://doi.org/10.1016/j.bbrc.2015.08.109>) improved background, but we still noted no difference in ubiquitin levels in wt versus *NogoA/B*^{-/-} cells. Thus, we are unsure whether *Ifitm3* indeed alters Nogo-B ubiquitination. We have removed all reference to this data in the manuscript. There are many ways in which *Ifitm3* may regulate Nogo-B in a proteasome-dependent manner and believe that detailed further detailed investigation of this is beyond the scope of the manuscript. However, we acknowledge this limitation of our study on page 17.

5. Are *Nogo-B/IFITM3* double knockout mice available for examination of their *in vivo* infection phenotype?

We have performed this experiment and show the data in new Fig 5. As predicted from our *in vitro* data, *Nogo-B* deficiency in *IFITM3*^{-/-} mice rescues virus-induced IL-6 (Fig. 5B) and weight loss (Fig. 5A) to levels measured in wt mice. Furthermore, we previously described that overt IL-6 in MCMV-infected *Ifitm3*^{-/-} mice leads to elevated virus load (Stacey et al, J Clin Invest., 2017). We now show that crossing *Ifitm3*^{-/-} mice to *Nogo-b*^{-/-} mice rescues this phenotype (Fig. 5C). We describe and discuss these data on pages 12-13. These data therefore highlight the critical importance of the *Ifitm3*-*Nogo-B* axis in regulating viral pathogenesis *in vivo*.

Minor comments:

1. Please check the accuracy of this statement in the introduction "...with reduced number of antiviral CD8+ T cells in IAV-infected lung tissue."

Thank you for noticing this. We have edited this sentence for improved accuracy.

2. Everitt, *Nature*, 2012, and Kenney, *PNAS*, 2019 could be added in reference to the following statement in the introduction as IL-6 and other cytokines were upregulated in tissues of *IFITM3* KO mice following influenza virus infection in these publications: "Similarly, in mouse models of viral infection, *Ifitm3* deficiency alters cytokine and chemokine profiles and leukocyte influx to sites of infection 19–21"

We have added these.

3. The following statement from the results is incorrect and the reference is cited inappropriately: "Although *IFITM3* has a negligible impact on SARS-CoV-2 cell entry 37." The cited publication shows that *IFITM3* can have distinct effects on SARS-CoV-2 entry in different cell types. Other publications have also suggested dual functions for *IFITM3* in SARS-CoV-2 infections, including Shi, *EMBO J*, 2021, and Winstone, *J Virol*, 2021.

Apologies for the lack of clarity. We have now improved this section including these key studies and highly relevant recent data linking *Ifitm3* to SARS-CoV-2-induced pathogenesis and pro-inflammatory cytokine production *in vivo*.

4. Can the authors speculate as to why other known interactors of *IFITM3* (e.g., *NEDD4*, *Fyn*, *AP-2*, *VAPA*, *ZMPSTE24*) were not pulled down in this study?

Numerous features can affect whether an IP pulls down a particular partner in any given experiment, including expression levels within the cell, interaction strength and longevity, the consequence of the interaction (e.g. if an interaction results in degradation, it is harder to detect), and the detection technology. In this study we used MS to detect binding partners, where both protein size (which affects the number of potential peptides produced), and inherent differences in ionisation efficiency, can affect detection – some proteins simply aren't easily detectable by MS. In addition, the majority of studies that detected the interactions listed by the reviewer used overexpression of both proteins, and pulled them down with high affinity epitope-tag specific antibodies. In contrast, our study performed IP of native endogenous (untagged) proteins. This is inherently a 'harder' experiment because it relies on a lower affinity antibody, binding to lower levels of target protein, and the antibody has the potential to interfere with some interactions. However, it has the significant advantage of revealing the interactions that are most relevant in the natural situation, without high level forced overexpression. What is critically relevant to our study is that the interaction between *Ifitm3* and *RTN4* has been independently validated in two separate studies, as highlighted on page 10.

5. Relevant comparisons in Fig 5B,D should be graphed next to each other. The current graph layouts are difficult to follow.

We have resolved this issue.

REVIEWERS' COMMENTS

Reviewer #1 (Remarks to the Author):

I am satisfied with the responses to my comments. This is a very nice study

Reviewer #2 (Remarks to the Author):

The authors have made a good faith effort in attempting to address all of the reviewer concerns. In particular, experiments testing double KO mice (IFITM3/NogoB) are an outstanding addition to the manuscript. These experiments convincingly support the conclusions of the paper with powerful genetic evidence. Overall, the authors have identified and characterized a novel function/mechanism of action for IFITM3 in regulating NogoB levels and TLR signaling to affect inflammation and outcomes of viral infection.

-Jacob Yount

Response to Reviewers' comments

Reviewer #1 (Remarks to the Author):

I am satisfied with the responses to my comments. This is a very nice study

We thank the reviewer for their appreciation of our study and for their helpful comments

Reviewer #2 (Remarks to the Author):

The authors have made a good faith effort in attempting to address all of the reviewer concerns. In particular, experiments testing double KO mice (IFITM3/NogoB) are an outstanding addition to the manuscript. These experiments convincingly support the conclusions of the paper with powerful genetic evidence. Overall, the authors have identified and characterized a novel function/mechanism of action for IFITM3 in regulating NogoB levels and TLR signaling to affect inflammation and outcomes of viral infection.

-Jacob Yount

We are grateful to the reviewer for their input into this study and for their appreciation of our findings.

Comment/Update for both reviewers:

We are grateful to both reviewers for their constructive input in the review process. We apologize for taking up more of your time, however we want to draw your attention/seek your approval for one additional change to Figure 7 that we have made during the final revision process:

When generating schematics of gating strategies of our flow cytometry data, as per Nature Communications policy (see new Supplementary Figure 6), we discovered that we could not find the original .fcs or .wsp files for the data in Figures 7d&e where we studied the impact of RTN4 and/or IFITM3 knockdown on surface TLR2 expression in THP1 cells. This error likely occurred when the postdoc who performed the experiment left the lab in Oxford after the initial manuscript submission. Subsequently, we only had data files for one replicate and felt that it was essential to repeat this prior to publication and replace our original data.

We therefore performed two additional repeats and both experiments reproduced the original findings. Our new data are now shown in Figure 7d&e, showing the same effect of RTN4 (Nogo-B) and/or IFITM3 knockdown on TLR2 expression, albeit with even greater rescue of surface TLR2 surface expression in double knockout cells, likely due to improved RTN4 siRNA protein knockdown (see page 14 for slight wording change). However, what we did notice with our repeat experiments (generated in Cardiff not Oxford with a different anti-TLR2 clone and flow cytometer) was that our TLR2 staining in these experiments (using the same THP1 cells as used previously) did not show clear separation of negative/positive

cells (Supplementary Figure 6b) as compared to our data in the original Fig 7d&e, and data from iPS-DCs (see Supplementary Figure 6a). Thus, we felt that the % values we originally used to express differences in TLR2 expression by THP1 cells was somewhat arbitrary and not representative of our THP1 dataset as a whole. Therefore, we now express data in Figure 7d as MFI to most accurately represent our results derived from this cell type.